# Streaming Heteroscedastic Probabilistic PCA with Missing Data

**Kyle Gilman**  *kgilman@umich.edu*
*Department of Electrical Engineering and Computer Science*
*University of Michigan*

**David Hong**  *hong@udel.edu*
*Department of Electrical and Computer Engineering*
*University of Delaware*

**Jeffrey A. Fessler**  *fessler@umich.edu*
*Department of Electrical Engineering and Computer Science*
*University of Michigan*

**Laura Balzano**  *girasole@umich.edu*
*Department of Electrical Engineering and Computer Science*
*University of Michigan*

**Reviewed on OpenReview:** *https://openreview.net/forum?id=lb2rPLuP9X*

## Abstract

Streaming principal component analysis (PCA) is an integral tool in large-scale machine learning for rapidly estimating low-dimensional subspaces from very high-dimensional data arriving at a high rate. However, modern datasets increasingly combine data from a variety of sources, and thus may exhibit heterogeneous quality across samples. Standard streaming PCA algorithms do not account for non-uniform noise, so their subspace estimates can quickly degrade. While the recently proposed Heteroscedastic Probabilistic PCA Technique (HePPCAT) addresses this heterogeneity, it was not designed to handle streaming data that may exhibit non-stationary behavior. Moreover, HePPCAT does not allow for missing entries in the data, which can be common in streaming data. This paper proposes the Streaming HeteroscedASTic Algorithm for PCA (SHASTA-PCA) to bridge this divide. SHASTA-PCA employs a stochastic alternating expectation-maximization approach that jointly learns the low-rank latent factors and the unknown noise variances from streaming data that may have missing entries and heteroscedastic noise, all while maintaining a low memory and computational footprint. Numerical experiments demonstrate the superior subspace estimation of our method compared to state-of-the-art streaming PCA algorithms in the heteroscedastic setting. Finally, we illustrate SHASTA-PCA applied to highly heterogeneous real data from astronomy.

## 1 Introduction

Modern data are increasingly large in scale and formed by combining heterogeneous samples from diverse sources or conditions that exhibit heteroscedastic noise, or noises of different variances (Hong et al., 2021),

Code for our project can be found at `https://github.com/kgilman/Streaming-Heteroscedastic-PPCA`.
K. Gilman & L. Balzano were supported in part by NSF CAREER award CCF-1845076 and AFOSR YIP award FA9550-19-1-0026. J.A. Fessler & L. Balzano were supported in part by NSF award CCF-2331590. D. Hong was supported by NSF Mathematical Sciences Postdoctoral Research Fellowship DMS 2103353.

such as in astronomy (Ahumada et al., 2020), medical imaging (Pruessmann et al., 1999; Anam et al., 2020), and cryo-electron microscopy imaging (Andén & Singer, 2017; Bendory et al., 2020). Principal component analysis (PCA) for visualization, exploratory data analysis, data compression, prediction, or other down-stream tasks is often a fundamental tool to process these high-dimensional data. However, several practical challenges arise when computing PCA on these types of data. In many applications, due to memory or physical constraints, the full data cannot be observed in their entirety at computation time and are instead read partially into memory piece by piece, or observations may stream in continuously and indefinitely. More-over, the signal may evolve over time and require adaptive tracking algorithms for the low-rank component. Adding to these difficulties, it is also common for big data to contain missing entries, such as in magnetic resonance imaging (Mensch et al., 2017), collaborative filtering (Candes & Plan, 2010), and environmental sensing (Ni et al., 2009). Consequently, there is a need for scalable streaming PCA techniques that can handle heteroscedastic noise and missing data.

A tremendous body of work has studied streaming PCA techniques for learning a signal subspace from noisy incremental data observations that have missing entries. Streaming or online PCA algorithms often enjoy the advantages of computational efficiency, low memory overhead, and adaptive tracking abilities, making them very useful in real-world big-data applications. However, no existing streaming methods account for noise with differing variances across samples, i.e., sample-wise heteroscedastic noise, and their subspace estimates can be highly corrupted by the noisiest samples. Hong et al. (2021) developed a Heteroscedastic Probabilistic PCA technique (HePPCAT) for data with varying noise levels across the samples. HePPCAT learns the low-rank factors and the unknown noise variances via maximum-likelihood estimation, but only in the batch setting with no missing entries. Other batch heteroscedastic PCA algorithms, like weighted PCA studied in Jolliffe (2002); Young (1941); Hong et al. (2023) and HeteroPCA (Zhang et al., 2022) for data with heteroscedastic features, also lack streaming and adaptive tracking abilities. None of the existing methods handle all of the real-data complexities we study here: missing entries, heteroscdastic noise, and streaming data. Tackling this non-trivial setting requires developing new algorithms.

To the best of our knowledge, this paper is the first work to develop a streaming PCA algorithm for data with missing entries and heteroscedastic noise. Our algorithm jointly estimates the factors and unknown noise variances in an *online* fashion from streaming incomplete data using an alternating stochastic minorize-maximize (SMM) approach with small computational and memory overhead. We carefully design minorizers with a particular alternating schedule of stochastic updates that distinguishes our approach from existing SMM methods. Notably, handling missing entries and heteroscedastic noise involves more complex updates than the simpler algebraic formulations of algorithms like HePPCAT or PETRELS (Chi et al., 2013). We demonstrate that our algorithm can estimate the signal subspace from subsampled data (even without know-ing the noise variances) better than state-of-the-art streaming PCA methods that assume homogeneous noise. Our algorithm is unique in that it not only tracks low-dimensional dynamic subspaces, but can also track dynamic noise variances that can occur, e.g., in sensor calibration (Jun-hua et al., 2003) and beamforming in nonstationary noise environments (Cohen, 2004). Finally, this work extends our understanding of streaming PCA to the setting of heteroscedastic noise and draws connections to existing work in the literature. In par-ticular, the proposed method closely relates to the streaming PCA algorithm PETRELS (Chi et al., 2013), but differs by learning the noise variances of the data on-the-fly and downweighting noisier data samples in the factor updates. Furthermore, we show that our proposed method implicitly optimizes a regularized least-squares problem whose adaptive hyperparameter varies by the learned heterogeneous noise variances.

Section 2 discusses related works for both streaming PCA and heteteroscedastic PCA. Sections 3 to 6 describe the model we consider, define the resulting optimization problem, and derive the proposed algorithm. Section 7 presents synthetic and real data experiments that demonstrate the benefits of the proposed method over existing state-of-the-art streaming PCA algorithms.

## 1.1 Notation

We use bold upper-case letters $\boldsymbol{A}$ to denote matrices, bold lower-case letters $\boldsymbol{v}$ to denote vectors, and non-bold lower-case letters $c$ for scalars. We denote the Hermitian transpose of a matrix as $\boldsymbol{A}'$ and the trace of a matrix as $\mathrm{tr}(\boldsymbol{A})$. The Euclidean norm is denoted by $\|\cdot\|_2$. The identity matrix of size $d \times d$ is denoted $\boldsymbol{I}_d$. The notation $i \in [k]$ means $i \in \{1, \ldots, k\}$.

## 2 Related work

### 2.1 Streaming PCA

A rich body of work has investigated a variety of streaming PCA algorithms for learning a signal subspace from incremental and possibly incomplete data observations. However, these methods assume that the data have homogeneous quality and do not model heteroscedastic properties like those considered in this paper. Since there are too many related works to detail here (see, e.g., Balzano et al. (2018); Boutsidis et al. (2016), for related references), we highlight a few of the most related.

One prominent branch of algorithms uses stochastic gradients to update the learned subspace based on a new data observation at each iteration; see, e.g., Bertsekas (2011) and Bottou (2010). Mardani et al. (2015) use stochastic gradient descent to learn matrix and tensor factorization models in the presence of missing data and also include an exponentially weighted data term that trades off adapting to new data with fitting historical data. Stochastic gradient descent over Riemannian manifolds is also a popular approach; see, e.g., Bonnabel (2013); Balzano et al. (2010); He et al. (2012); Goes et al. (2014). Oja's method (Oja, 1982) takes a stochastic gradient step to update the subspace basis from the most recent data vector, obtaining a new orthonormal basis after orthogonalization. Balzano (2022) proved the equivalence of the GROUSE algorithm (Balzano et al., 2010) with Oja's method for a certain step size. Other progress on improving and understanding Oja's method has recently been made, such as an algorithm to adaptively select the learning rate in one pass over the data (Henriksen & Ward, 2019) and an analysis of the convergence rate for non-i.i.d. data sampled from a Markov chain (Kumar & Sarkar, 2024).

Some streaming PCA methods share commonalities with quasi-second-order optimization methods. For example, the PETRELS algorithm proposed in Chi et al. (2013) fits a factor model to data with missing entries via a stochastic quasi-Newton method. PETRELS has computationally efficient updates but can encounter numerical instability issues in practice after a large number of samples.

More recently, streaming PCA and its analogs have been extended to a variety of new problems. Giannakis et al. (2023) propose a streaming algorithm for forecasting dynamical systems that they show is a type of streaming PCA problem. Streaming algorithms have been proposed for robust PCA (Thanh et al., 2021; Diakonikolas et al., 2023), federated learning and differential privacy (Grammenos et al., 2020), a distributed Krasulina's method (Raja & Bajwa, 2022), and probabilistic PCA to track nonstationary processes (Lu et al., 2024). Although not strictly a streaming algorithm, Blocker et al. (2023) also estimate dynamic subspaces but by using a piecewise-geodesic model on the Grassmann manifold. All of these approaches implicitly assume that the data quality is homogeneous across the dataset, in contrast to our proposed approach.

### 2.2 Stochastic MM methods

Another vein of work on streaming algorithms, which has the closest similarities to this paper, is stochastic majorization-minimization (SMM) algorithms for matrix and tensor factorization. MM methods construct surrogate functions that are more easily optimized than the original objective; algorithms that successively optimize these surrogates are guaranteed to converge to stationary points of the original objective function under suitable regularity conditions (Jacobson & Fessler, 2007; Lange, 2016). SMM algorithms such as Mairal (2013); Lyu et al. (2022); Lyu (2024) update and optimize a stochastic approximation of the surrogate function upon observing each new data sample. The work in Mairal (2013) proved almost sure convergence to a stationary point for non-convex objectives with one block of variables for the SMM technique. The work in Mensch et al. (2017) proposes subsampled online matrix factorization (SOMF) for large-scale streaming subsampled data and gives convergence guarantees under mild assumptions. In Lyu et al. (2022); Lyu (2024), the authors extend SMM to functions that are multi-convex in blocks of variables for online tensor factorization. Their framework performs block-coordinate minimization of a single majorizer at each time point. They prove almost sure convergence of the iterates to a stationary point assuming certain regularity conditions of the loss function and learning rate and that the data tensors follow a Markovian process. and the data tensors form a Markov chain with a unique stationary distribution. More recently, Phan et al. (2024) propose and analyze several stochastic variance-reduced MM algorithms. Like these works, our paper also draws upon SMM techniques; we use an alternating SMM approach to optimize the log-likelihood function

for our model. However, our setting and approach differ from these existing methods in key ways; we detail these differences and their impact on the related convergence theory in §6.3.1.

A close analogue to SMM is the Doubly Stochastic Successive Convex (DSSC) approximation algorithm (Mokhtari & Koppel, 2020) that optimizes convex surrogates to non-convex objective functions from streaming samples or minibatches. A key feature of their algorithm is that it decomposes the optimization variable into $B$ blocks and operates on random subsets of blocks at each iteration. Specifically, the DSSC algorithm chooses a block $i \in [B]$, computes stochastic gradients with respect to the $i$th block of variables and then recursively updates the approximation to the $i$th surrogate function. From the optimizer to the approximate surrogate, their algorithm performs momentum updates of the iterates very similarly to SMM. Our own algorithm SHASTA-PCA in §6 can be interpreted as following a similar approach, but without using gradient methods since our problem does not have Lipschitz-continuous gradients.

## 2.3 Heterogeneous data

Several recently proposed PCA algorithms consider data contaminated by heteroscedastic noise *across samples*, which is the setting we study in this paper. Weighted PCA is a natural approach in this context (Jolliffe, 2002), either weighting the samples by the inverse noise variances (Young, 1941) or by an optimal weighting derived in Hong et al. (2023). In both instances, the variances must be known *a priori* or estimated to compute the weights. Probabilistic PCA (PPCA) (Tipping & Bishop, 1999b) uses a probabilistic interpretation of PCA via a factor analysis model with isotropic Gaussian noise and latent variables. For a single unknown noise variance (i.e., homoscedastic noise), the learned factors and noise variance are solutions to a maximum-likelihood problem that can be optimized using an expectation maximization algorithm; these solutions correspond exactly to PCA. Hong et al. (2021) studied the heteroscedastic probabilistic PCA (HPPCA) problem that considers a factor model where groups of data may have different (unknown) noise variances. Their method, HePPCAT, performs maximum-likelihood estimation of the latent factors and unknown noise variances (assuming knowledge of which samples belong to each noise variance group); they consider various algorithms and recommend an alternating EM approach. Other batch heteroscedastic PCA methods have since followed HePPCAT. ALPACAH (Salazar Cavazos et al., 2025) estimates the low-rank component and variances for data with sample-wise heteroscedastic noise, but no streaming counterpart currently exists. HeMPPCAT (Xu et al., 2023) extends mixtures of probabilistic PCA (Tipping & Bishop, 1999a) to the case of heteroscedastic noise across samples.

More broadly, there is an increasing body of work that investigates PCA techniques for data contaminated by some sort of heterogeneous noise, including noise that is heteroscedastic *across features*. HeteroPCA (Zhang et al., 2022) iteratively imputes the diagonal entries of the sample covariance matrix to address bias that arises when the noise has feature-wise heteroscedasticity, and Zhou & Chen (2025) then extended HeteroPCA to the case of ill-conditioned low-rank data. Another line of work in Leeb & Romanov (2021) and Leeb (2021) has considered rescaling the data to instead whiten the noise. Yan et al. (2024) develops inference and uncertainty quantification procedures for PCA with missing data and feature-wise heteroscedasticity. There has also been recent progress on developing methods to estimate the rank in heterogeneous noise contexts (Hong et al., 2020; Ke et al., 2021; Landa et al., 2022; Landa & Kluger, 2025) and on establishing fundamental limits for recovery in these settings (Behne & Reeves, 2022; Zhang & Mondelli, 2024).

A closely related problem in signal processing applications, such as heterogeneous clutter in radar, is data with heterogeneous "textures", also called the "mixed effects" problem. Here, the signal is modeled as a mixture of scaled Gaussians, each sharing a common low-rank covariance scaled by an unknown deterministic positive "texture" or power factor (Breloy et al., 2019). In fact, the heterogeneous texture and HPPCA problems are related up to an unknown scaling (Hong et al., 2021). Ferrer et al. (2021) and Hippert-Ferrer et al. (2022) also studied variations of the heterogeneous texture problem for robust covariance matrix estimation from batch data with missing entries. Collas et al. (2021) study the probabilistic PCA problem in the context of isotropic signals with unknown heterogeneous textures and a known noise floor. Their paper casts the maximum-likelihood estimation as an optimization problem over a Riemannian manifold, using gradient descent on the manifold to jointly optimize for the subspace and the textures. Their formulation also readily admits a stochastic gradient algorithm for online optimization.

Heteroscedastic data has also been investigated in the setting of supervised learning for fitting linear regression models with stochastic gradient descent (Song et al., 2015). The authors show that the model's performance given "clean" and "noisy" datasets depends on the learning rate and the order in which the datasets are processed. Further, they propose using separate learning rates that depend on the noise levels instead of using one learning rate as is done in classical SGD. In the context of generalized linear bandits, Zhao et al. (2023) propose an online algorithm for the heteroscedastic bandit problem using weighted linear regression with weights selected as the inverse noise variance.

## 3 Probabilistic Model

Similar to Hong et al. (2019; 2021), we model data samples in $\mathbb{R}^d$ from $L$ noise level groups as:

$$\boldsymbol{y}_i = \boldsymbol{F}\boldsymbol{z}_i + \boldsymbol{\varepsilon}_i, \qquad \text{for } i = 1, 2, \ldots, \tag{1}$$

where $\boldsymbol{F} \in \mathbb{R}^{d \times k}$ is a deterministic factor matrix to estimate, $\boldsymbol{z}_i \sim \mathcal{N}(\boldsymbol{0}_k, \boldsymbol{I}_k)$ are independent and identically distributed (i.i.d.) coefficient vectors, $\boldsymbol{\varepsilon}_i \sim \mathcal{N}(\boldsymbol{0}_d, v_{g_i}\boldsymbol{I}_d)$ are independent noise vectors, $g_i \in \{1, \ldots, L\}$ is the noise level group to which the $i$th sample belongs, and $v_1, \ldots, v_L$ are deterministic noise variances to estimate. Typically, we assume $k \ll d$ to model data from a low-dimensional subspace. We also assume the group memberships $g_i$ are known. This is a valid assumption in some real applications where we know the sources of the data, e.g., the sensor type in air quality monitoring (Hong et al., 2021), or data from a particular coil in the MRI machine.

Let $\Omega_i \subseteq \{1, \ldots, d\}$ denote the set of entries observed for the $i$th sample, and let $\boldsymbol{y}_{\Omega_i} \in \mathbb{R}^{|\Omega_i|}$ and $\boldsymbol{F}_{\Omega_i} \in \mathbb{R}^{|\Omega_i| \times k}$ denote the restrictions of $\boldsymbol{y}_i$ and $\boldsymbol{F}$ to the entries and rows defined by $\Omega_i$. Then the observed entries of the data vectors are distributed as

$$\boldsymbol{y}_{\Omega_i} \sim \mathcal{N}(\boldsymbol{0}_{|\Omega_i|}, \boldsymbol{F}_{\Omega_i}\boldsymbol{F}'_{\Omega_i} + v_{g_i}\boldsymbol{I}_{|\Omega_i|}).$$

We will express the joint log-likelihood over only the *observed* entries of the data and maximize it for the unknown deterministic model parameters.

For a batch of $n$ vectors, the joint log-likelihood over the observed batch data for $\Omega = (\Omega_1, \ldots, \Omega_n)$ can be easily written in an incremental form as a sum of log-likelihoods over the partially observed dataset $\boldsymbol{Y}_\Omega \triangleq (\boldsymbol{y}_{\Omega_1}, \ldots, \boldsymbol{y}_{\Omega_n})$:

$$\mathcal{L}(\boldsymbol{Y}_\Omega; \boldsymbol{F}, \boldsymbol{v}) = \frac{1}{2}\sum_{i=1}^n \mathcal{L}_i(\boldsymbol{y}_{\Omega_i}; \boldsymbol{F}, \boldsymbol{v}) + C, \tag{2}$$

where $\boldsymbol{v} = [v_1, \ldots, v_L]'$, $C$ is a constant independent of $\boldsymbol{F}$ and $\boldsymbol{v}$, and

$$\mathcal{L}_i(\boldsymbol{y}_{\Omega_i}; \boldsymbol{F}, \boldsymbol{v}) \triangleq \ln\det(\boldsymbol{F}_{\Omega_i}\boldsymbol{F}'_{\Omega_i} + v_{g_i}\boldsymbol{I}_{|\Omega_i|})^{-1} - \boldsymbol{y}'_{\Omega_i}(\boldsymbol{F}_{\Omega_i}\boldsymbol{F}'_{\Omega_i} + v_{g_i}\boldsymbol{I}_{|\Omega_i|})^{-1}\boldsymbol{y}_{\Omega_i} \tag{3}$$

is the loss for a single vector $\boldsymbol{y}_{\Omega_i}$. To jointly estimate the factor matrix $\boldsymbol{F}$ and the variances $\boldsymbol{v}$, we maximize this likelihood. Optimizing the log-likelihood (2) is a challenging non-concave optimization problem, so we propose an efficient alternating minorize-maximize (MM) approach.

## 4 Expectation Maximization Minorizer

This section derives a minorizer for the log-likelihood (2) that will be used to develop the proposed alternating MM algorithm in the following sections. In particular, we derive a minorizer at the point $(\widetilde{\boldsymbol{F}}, \widetilde{\boldsymbol{v}})$ in the style of expectation maximization methods. The minorizer follows from the work in Hong et al. (2021); here we extend it to the case for data with missing entries.

For the complete-data log-likelihood, we use the observed samples $\boldsymbol{y}_{\Omega_i}$ and unknown coefficients $\boldsymbol{z}_i$, leading to the following complete-data log-likelihood for the $i$th sample:

$$
\begin{aligned}
\mathcal{L}_i^c(\boldsymbol{F}, \boldsymbol{v}) &\triangleq \ln p(\boldsymbol{y}_{\Omega_i}, \boldsymbol{z}_i; \boldsymbol{F}, \boldsymbol{v}) \\
&= \ln p(\boldsymbol{y}_{\Omega_i} | \boldsymbol{z}_i; \boldsymbol{F}, \boldsymbol{v}) + \ln p(\boldsymbol{z}_i; \boldsymbol{F}, \boldsymbol{v}) \\
&= -\frac{|\Omega_i|}{2} \ln v_{g_i} - \frac{\|\boldsymbol{y}_{\Omega_i} - \boldsymbol{F}_{\Omega_i} \boldsymbol{z}_i\|_2^2}{2v_{g_i}} - \frac{\|\boldsymbol{z}_i\|_2^2}{2},
\end{aligned}
\tag{4}
$$

where (4) drops the constants $\ln(2\pi)^{-|\Omega_i|/2}$ and $\ln(2\pi)^{-k/2}$.

Next, we take the expectation of (4) with respect to the following conditional distribution $\boldsymbol{z}|\boldsymbol{y}$, derived in Appendix B:

$$
\boldsymbol{z}_i | \{\boldsymbol{y}_{\Omega_i}, \boldsymbol{F} = \widetilde{\boldsymbol{F}}, \boldsymbol{v} = \tilde{\boldsymbol{v}}\} \overset{\text{ind}}{\sim} \mathcal{N}(\check{\boldsymbol{z}}_i(\widetilde{\boldsymbol{F}}, \tilde{\boldsymbol{v}}), \tilde{v}_{g_i} \boldsymbol{M}_i(\widetilde{\boldsymbol{F}}, \tilde{\boldsymbol{v}})),
$$

where we define $\check{\boldsymbol{z}}_i$ and $\boldsymbol{M}_i$ for use here and in following derivations as:

$$
\check{\boldsymbol{z}}_i(\boldsymbol{F}, \boldsymbol{v}) \triangleq \boldsymbol{M}_i(\boldsymbol{F}, \boldsymbol{v}) \boldsymbol{F}_{\Omega_i}' \boldsymbol{y}_{\Omega_i}
\tag{5}
$$

$$
\boldsymbol{M}_i(\boldsymbol{F}, \boldsymbol{v}) \triangleq (\boldsymbol{F}_{\Omega_i}' \boldsymbol{F}_{\Omega_i} + v_{g_i} \boldsymbol{I}_k)^{-1}.
\tag{6}
$$

Doing so yields the following minorizer for $\mathcal{L}_i$ at $(\widetilde{\boldsymbol{F}}, \tilde{\boldsymbol{v}})$:

$$
\begin{aligned}
\Psi_i(\boldsymbol{F}, \boldsymbol{v}; \widetilde{\boldsymbol{F}}, \tilde{\boldsymbol{v}}) &\triangleq -\frac{|\Omega_i|}{2} \ln v_{g_i} - \frac{\|\boldsymbol{y}_{\Omega_i}\|_2^2}{2v_{g_i}} + \frac{1}{v_{g_i}} \boldsymbol{y}_{\Omega_i}' \boldsymbol{F}_{\Omega_i} \check{\boldsymbol{z}}_i(\widetilde{\boldsymbol{F}}, \tilde{\boldsymbol{v}}) \\
&\quad - \frac{1}{2v_{g_i}} \left( \|\boldsymbol{F}_{\Omega_i} \check{\boldsymbol{z}}_i(\widetilde{\boldsymbol{F}}, \tilde{\boldsymbol{v}})\|_2^2 + \tilde{v}_{g_i} \operatorname{tr}\{\boldsymbol{F}_{\Omega_i}' \boldsymbol{F}_{\Omega_i} \boldsymbol{M}_i(\widetilde{\boldsymbol{F}}, \tilde{\boldsymbol{v}})\} \right),
\end{aligned}
\tag{7}
$$

where (7) drops terms that are constant with respect to $\boldsymbol{F}$ and $\boldsymbol{v}$.

## 5  A Batch Algorithm

Before deriving the proposed streaming algorithm, SHASTA-PCA, we first derive a batch method for comparison purposes. Summing the sample-wise minorizer (7) across all the samples gives the following batch minorizer at the point $(\widetilde{\boldsymbol{F}}, \tilde{\boldsymbol{v}})$:

$$
\begin{aligned}
\Psi(\boldsymbol{F}, \boldsymbol{v}; \widetilde{\boldsymbol{F}}, \tilde{\boldsymbol{v}}) &\triangleq \sum_{i=1}^n \Psi_i(\boldsymbol{F}, \boldsymbol{v}; \widetilde{\boldsymbol{F}}, \tilde{\boldsymbol{v}}) \\
&= \sum_{\ell=1}^L \sum_{i : g_i = \ell} -\frac{|\Omega_i|}{2} \ln v_\ell - \frac{\|\boldsymbol{y}_{\Omega_i}\|_2^2}{2v_\ell} + \frac{1}{v_\ell} \boldsymbol{y}_{\Omega_i}' \boldsymbol{F}_{\Omega_i} \check{\boldsymbol{z}}_i(\widetilde{\boldsymbol{F}}, \tilde{\boldsymbol{v}}) \\
&\quad - \frac{1}{2v_\ell} \left( \|\boldsymbol{F}_{\Omega_i} \check{\boldsymbol{z}}_i(\widetilde{\boldsymbol{F}}, \tilde{\boldsymbol{v}})\|_2^2 + \tilde{v}_\ell \operatorname{tr}\{\boldsymbol{F}_{\Omega_i}' \boldsymbol{F}_{\Omega_i} \boldsymbol{M}_i(\widetilde{\boldsymbol{F}}, \tilde{\boldsymbol{v}})\} \right).
\end{aligned}
\tag{8}
$$

Similar to HePPCAT (Hong et al., 2021), which is a batch method for fully sampled data, in each iteration $t$, we first update $\boldsymbol{v}$ (with $\boldsymbol{F}$ fixed at $\boldsymbol{F}_{t-1}$) then update $\boldsymbol{F}$ (with $\boldsymbol{v}$ fixed at $\boldsymbol{v}_t$), i.e.,

$$
\boldsymbol{v}_t = \arg\max_{\boldsymbol{v}} \Psi(\boldsymbol{F}_{t-1}, \boldsymbol{v}; \boldsymbol{F}_{t-1}, \boldsymbol{v}_{t-1}),
\tag{9}
$$

$$
\boldsymbol{F}_t = \arg\max_{\boldsymbol{F}} \Psi(\boldsymbol{F}, \boldsymbol{v}_t; \boldsymbol{F}_{t-1}, \boldsymbol{v}_t).
\tag{10}
$$

Here $i$ refers to index of the data sample, and $t$ denotes only the algorithm iteration, in contrast to the streaming algorithm in §6, where $t$ denotes both the time index (sample) and algorithm iteration. The following subsections derive efficient formulas for these updates and discuss the memory and computational costs.

### 5.1 Optimizing $v$ for fixed $F$

Here we derive an efficient formula for the $v$ update in (9). While the update is similar to HePPCAT (Hong et al., 2021), the key difference lies in computing the minorizer parameters. Specifically, the missing data makes the sample-wise quantities $\check{z}_i(\cdot, \cdot)$ and $M_i(\cdot, \cdot)$ in (5) and (6) depend on the sampling patterns for the $i$th data vector and must be computed for every sample $i \in [n]$ per iteration compared to the single $\check{z}$ and $\widetilde{M}$ used in HePPCAT. This update separates into $L$ univariate optimization problems, one in each variance $v_\ell$:

$$v_{t,\ell} = \arg\max_{v_\ell} -\frac{\theta^{\text{batch}}_{t,\ell}}{2}\ln v_\ell - \frac{\rho^{\text{batch}}_{t,\ell}}{2v_\ell},$$

where

$$\theta^{\text{batch}}_{t,\ell} \triangleq \sum_{i\,:\,g_i=\ell} |\Omega_i|,$$

$$\rho^{\text{batch}}_{t,\ell} \triangleq \sum_{i\,:\,g_i=\ell} \left[ \|\boldsymbol{y}_{\Omega_i} - \boldsymbol{F}_{t-1,\Omega_i}\check{z}_i(\boldsymbol{F}_{t-1}, \boldsymbol{v}_{t-1})\|_2^2 + v_{t-1,\ell}\,\text{tr}(\boldsymbol{F}'_{t-1,\Omega_i}\boldsymbol{F}_{t-1,\Omega_i}\boldsymbol{M}_i(\boldsymbol{F}_{t-1}, \boldsymbol{v}_{t-1})) \right],$$

$\boldsymbol{F}_{t-1,\Omega_i}$ denotes the iterate $\boldsymbol{F}_{t-1}$ restricted to the rows defined by $\Omega_i$. The corresponding solutions are

$$v_{t,\ell} = \frac{\rho^{\text{batch}}_{t,\ell}}{\theta^{\text{batch}}_{t,\ell}}.$$

We precompute $\theta^{\text{batch}}_{t,\ell}$ because it remains constant across iterations.

### 5.2 Optimizing $F$ for fixed $v$

Here we derive an efficient formula for the $F$ update in (10). The update differs from the factor update in HePPCAT again due to the missing entries in the data. Specifically, this update separates into $d$ quadratic optimization problems, one in each row $f_j$ of $F$:

$$\boldsymbol{f}_{t,j} = \arg\max_{\boldsymbol{f}_j} \boldsymbol{f}'_j \boldsymbol{s}^{\text{batch}}_{t,j} - \frac{1}{2}\boldsymbol{f}'_j \boldsymbol{R}^{\text{batch}}_{t,j}\boldsymbol{f}_j,$$

where

$$\boldsymbol{R}^{\text{batch}}_{t,j} \triangleq \sum_{\ell=1}^{L} \sum_{\substack{i\,:\,g_i=\ell \\ \Omega_i \ni j}} \frac{1}{v_{t,\ell}}(\check{z}_i(\boldsymbol{F}_{t-1}, \boldsymbol{v}_t)\check{z}_i(\boldsymbol{F}_{t-1}, \boldsymbol{v}_t)' + v_{t,\ell}\boldsymbol{M}_i(\boldsymbol{F}_{t-1}, \boldsymbol{v}_t))$$

$$\boldsymbol{s}^{\text{batch}}_{t,j} \triangleq \sum_{\ell=1}^{L} \sum_{\substack{i\,:\,g_i=\ell \\ \Omega_i \ni j}} \frac{1}{v_{t,\ell}}y_{ij}\check{z}_i(\boldsymbol{F}_{t-1}, \boldsymbol{v}_t),$$

capture the data-dependent terms in the minorizer in (8), and $y_{ij}$ is the $j$th coordinate of the vector $\boldsymbol{y}_i$. We compute the solutions for $\boldsymbol{f}_j$ in parallel as

$$\boldsymbol{f}_{t,j} = \left(\boldsymbol{R}^{\text{batch}}_{t,j}\right)^{-1}\boldsymbol{s}^{\text{batch}}_{t,j} \quad \forall j \in [d].$$

### 5.3 Memory and Computational Complexity

The batch algorithm above involves first accessing all $n = \sum_{\ell=1}^{L} n_\ell$ data vectors to compute the minorizer parameters $\boldsymbol{M}_i(\cdot, \cdot) \in \mathbb{R}^{k \times k}$ and $\check{z}_i(\cdot, \cdot) \in \mathbb{R}^k$ at a cost of $\mathcal{O}(nk^3 + \sum_{\ell=1}^{L}\sum_{i\,:\,g_i=\ell}|\Omega_i|k^2)$ flops per iteration and $\mathcal{O}(n(k^2+k))$ memory elements. Computing $\boldsymbol{R}^{\text{batch}}_{t,j}$ and $\boldsymbol{s}^{\text{batch}}_{t,j}$ incurs a cost of $\mathcal{O}(\sum_{\ell=1}^{L}\sum_{i\,:\,g_i=\ell}|\Omega_i|(k^2+k))$ flops for all $j = 1, ..., d$, and finally solving for the rows of $F$ costs $\mathcal{O}(dk^3)$ flops per iteration. Updating $v$ requires $\mathcal{O}(\sum_{\ell=1}^{L}\sum_{i\,:\,g_i=\ell}|\Omega_i|k^2)$ computations.

Since each complete update depends on all $n$ samples, the batch algorithm must have access to the entire dataset at run-time, either by reading over all the data in multiple passes while accumulating the computed terms used to parameterize the minorizers, or by storing all the data at once, which requires $\mathcal{O}(\sum_{i=1}^{n} |\Omega_i| + dk^2)$ memory. This requirement, combined with the $\mathcal{O}(n)$ inversions of $k \times k$ matrices in each iteration, significantly limits the practicality of the batch algorithm for massive-scale or high-arrival-rate data as well as in infinite-streaming applications.

## 6 Proposed Algorithm: SHASTA-PCA

The structure of the log-likelihood in (2) suggests a natural way to perform incremental (in the finite data setting) or stochastic (in expectation) updates. If each data sample from the $\ell$th group is drawn i.i.d. from the model in (1), then under uniform random sampling of the data entries, each $\mathcal{L}_i$ is an unbiased estimator of $\mathcal{L}$. Hence, we leverage the work in Mairal (2013), which proposed a stochastic MM (SMM) technique for optimizing empirical loss functions from large-scale or possibly infinite data sets. For the loss function we consider, these online algorithms have recursive updates with a light memory footprint that is independent of the number of samples.

In the streaming setting with only a single observation $\boldsymbol{y}_{\Omega_t}$ at each time index $t$, we do not have access to the full batch minorizer in (8), but rather only a single $\Psi_t(\boldsymbol{F}, \boldsymbol{v}; \widetilde{\boldsymbol{F}}, \widetilde{\boldsymbol{v}})$. Key to our approach, for each $t$, our proposed algorithm uses $\Psi_t(\boldsymbol{F}, \boldsymbol{v}; \widetilde{\boldsymbol{F}}, \widetilde{\boldsymbol{v}})$ to update two separate approximations to $\Psi(\boldsymbol{F}, \boldsymbol{v}; \widetilde{\boldsymbol{F}}, \widetilde{\boldsymbol{v}})$ parameterized by $\boldsymbol{F}$ and $\boldsymbol{v}$, respectively, i.e., $\bar{\Psi}_t^{(F)}(\boldsymbol{F})$ and $\bar{\Psi}_t^{(v)}(\boldsymbol{v})$, in an alternating way. While other optimization approaches are possible—for example, performing block coordinate maximization of a single joint approximate majorizer—our novel approach of alternating between the two separate approximate minorizers reduces memory usage and computational overhead in our setting. Given a sequence of non-increasing positive weights $\{w_t\}_{t \geq 0} \in (0, 1)$ and positive scalars $c_v$ and $c_F$, we first update $\boldsymbol{v}$ (with $\boldsymbol{F}$ fixed at $\boldsymbol{F}_{t-1}$) with one SMM iteration, then update $\boldsymbol{F}$ (with $\boldsymbol{v}$ fixed at $\boldsymbol{v}_t$) with another. Namely, we have the noise variance update:

$$\bar{\Psi}_t^{(v)}(\boldsymbol{v}) = (1 - w_t)\bar{\Psi}_{t-1}^{(v)}(\boldsymbol{v}) + w_t \Psi_t(\boldsymbol{F}_{t-1}, \boldsymbol{v}; \boldsymbol{F}_{t-1}, \boldsymbol{v}_{t-1}), \tag{11}$$

$$\boldsymbol{v}_t = (1 - c_v)\boldsymbol{v}_{t-1} + c_v \arg\max_{\boldsymbol{v}} \bar{\Psi}_t^{(v)}(\boldsymbol{v}), \tag{12}$$

followed by the factor update:

$$\bar{\Psi}_t^{(F)}(\boldsymbol{F}) = (1 - w_t)\bar{\Psi}_{t-1}^{(F)}(\boldsymbol{F}) + w_t \Psi_t(\boldsymbol{F}, \boldsymbol{v}_t; \boldsymbol{F}_{t-1}, \boldsymbol{v}_t) \tag{13}$$

$$\boldsymbol{F}_t = (1 - c_F)\boldsymbol{F}_{t-1} + c_F \arg\max_{\boldsymbol{F}} \bar{\Psi}_t^{(F)}(\boldsymbol{F}). \tag{14}$$

The iterate averaging updates in (12) and (14) are important to control the distance between iterates and have both practical and theoretical significance in SMM algorithms (Lyu et al., 2022; Lyu, 2024). Empirically, we found that using constant $c_F$ and $c_v$ worked well, but other iterate averaging techniques are also possible, such as those discussed in Mairal (2013). Other ways to control the iterates include optimizing over a trust region, as done in Lyu et al. (2022).

Since the iterate and the time index are the same in the streaming setting, i.e., $t = i$, we now denote both the sample and the SMM iteration by $t$ in the remainder of this section. We now derive efficient recursive updates and compare the memory and computational costs to the batch algorithm.

### 6.1 Optimizing $v$ for fixed $F$

Now note that the minorizer in (7) for $i = t$ and $(\widetilde{\boldsymbol{F}}, \widetilde{\boldsymbol{v}}) = (\boldsymbol{F}_{t-1}, \boldsymbol{v}_{t-1})$ can be re-written as

$$\Psi_t(\boldsymbol{F}_{t-1}, \boldsymbol{v}; \boldsymbol{F}_{t-1}, \boldsymbol{v}_{t-1}) = C_t - |\Omega_t| \ln v_{g_t} - \frac{\rho_t}{v_{g_t}},$$

where $C_t$ does not depend on $\boldsymbol{v}$ and

$$\rho_t \triangleq \|\boldsymbol{y}_{\Omega_t} - \boldsymbol{F}_{t-1,\Omega_t}\check{\boldsymbol{z}}_t(\boldsymbol{F}_{t-1}, \boldsymbol{v}_{t-1})\|_2^2 + v_{t-1,g_t} \operatorname{tr}(\boldsymbol{F}'_{t-1,\Omega_t}\boldsymbol{F}_{t-1,\Omega_t}\boldsymbol{M}_t(\boldsymbol{F}_{t-1}, \boldsymbol{v}_{t-1})).$$

Recall that $g_t \in [L]$ is the group index of the $t$th data vector. Thus, it follows from Equation (11) that

$$\bar{\Psi}_t^{(v)}(\boldsymbol{v}) = C_t' + \sum_{\ell=1}^{L} -\bar{\theta}_{t,\ell} \ln v_\ell - \frac{\bar{\rho}_{t,\ell}}{v_\ell},$$

where $C_t'$ is a constant that does not depend on $\boldsymbol{v}$,

$$\bar{\theta}_{t,g_t} \triangleq (1 - w_t)\bar{\theta}_{t-1,g_t} + w_t|\Omega_t|, \tag{15}$$

$$\bar{\rho}_{t,g_t} \triangleq (1 - w_t)\bar{\rho}_{t-1,g_t} + w_t\rho_t, \tag{16}$$

and for $\ell \neq g_t$

$$\bar{\theta}_{t,\ell} \triangleq (1 - w_t)\bar{\theta}_{t-1,\ell}, \quad \bar{\rho}_{t,\ell} \triangleq (1 - w_t)\bar{\rho}_{t-1,\ell}. \tag{17}$$

The $\ell$th term in the sum is optimized by $v_\ell = \bar{\rho}_{t,\ell}/\bar{\theta}_{t,\ell}$, so

$$v_{t,\ell} = (1 - c_v)v_{t-1,\ell} + c_v\frac{\bar{\rho}_{t,\ell}}{\bar{\theta}_{t,\ell}}. \tag{18}$$

Here the vectors $\bar{\boldsymbol{\theta}}_t \in \mathbb{R}^L$ and $\bar{\boldsymbol{\rho}}_t \in \mathbb{R}^L$ aggregate past information to parameterize the approximate minorizer in $\boldsymbol{v}$. Since there is no past information at $t = 0$, we chose to initialize them with zero vectors. However, in the initial iterations where no data vectors have been observed for the $\ell$th group, (18) is undefined, so a valid argument maximizing (12) is simply $v_{0,\ell}$, i.e., the initialized value.

## 6.2 Optimizing $\boldsymbol{F}$ for fixed $\boldsymbol{v}$

Maximizing the approximate minorizer $\bar{\Psi}_t^{(F)}(\boldsymbol{F})$ with respect to $\boldsymbol{F}$ reduces to maximizing $d$ quadratics in the rows of $\boldsymbol{F}$ for $j \in [d]$:

$$\bar{\Psi}_t^{(F)}(\boldsymbol{F}) = \sum_{j=1}^{d} \boldsymbol{f}_j'\bar{\boldsymbol{s}}_{t,j} - \boldsymbol{f}_j'\overline{\boldsymbol{R}}_{t,j}\boldsymbol{f}_j, \tag{19}$$

where for $j \in \Omega_t$

$$\overline{\boldsymbol{R}}_{t,j} = (1 - w_t)\overline{\boldsymbol{R}}_{t-1,j} + w_t\boldsymbol{R}_{t,j}, \tag{20}$$

$$\bar{\boldsymbol{s}}_{t,j} = (1 - w_t)\bar{\boldsymbol{s}}_{t-1,j} + w_t\boldsymbol{s}_{t,j}, \tag{21}$$

where

$$\boldsymbol{R}_{t,j} \triangleq \frac{1}{2}\left(\frac{1}{v_{t,g_t}}\check{\boldsymbol{z}}_t(\boldsymbol{F}_{t-1}, \boldsymbol{v}_t)\check{\boldsymbol{z}}_t(\boldsymbol{F}_{t-1}, \boldsymbol{v}_t)' + \boldsymbol{M}_t(\boldsymbol{F}_{t-1}, \boldsymbol{v}_t)\right), \quad \boldsymbol{s}_{t,j} \triangleq \frac{1}{v_{t,g_t}}y_{tj}\check{\boldsymbol{z}}_t(\boldsymbol{F}_{t-1}, \boldsymbol{v}_t), \tag{22}$$

and for $j \notin \Omega_t$

$$\overline{\boldsymbol{R}}_{t,j} = (1 - w_t)\overline{\boldsymbol{R}}_{t-1,j}, \quad \bar{\boldsymbol{s}}_{t,j} = (1 - w_t)\bar{\boldsymbol{s}}_{t-1,j}. \tag{23}$$

The parameters $(\overline{\boldsymbol{R}}_{t,j}, \bar{\boldsymbol{s}}_{t,j})$ for $j \in [d]$ of the approximate minorizer $\bar{\Psi}_t^{(F)}(\boldsymbol{F})$ aggregate past information from previously observed samples, permitting our algorithm to stream over an arbitrary amount of data while using a constant amount of memory.

Maximizing the approximate minorizer $\bar{\Psi}_t^{(F)}(\boldsymbol{F})$ with respect to each row of $\boldsymbol{F}$ yields

$$\hat{\boldsymbol{f}}_j = \overline{\boldsymbol{R}}_{t,j}^{-1}\bar{\boldsymbol{s}}_{t,j}, \quad i = 1, \ldots, d. \tag{24}$$

Since the problem separates in each row of $\boldsymbol{F}$, this form permits efficient parallel computations. Further, because $\hat{\boldsymbol{f}}_j = \overline{\boldsymbol{R}}_{t,j}^{-1}\bar{\boldsymbol{s}}_{t,j} = \overline{\boldsymbol{R}}_{t-1,j}^{-1}\bar{\boldsymbol{s}}_{t-1,j}$ for $j \notin \Omega_t$, we solve the $k \times k$ linear systems in (24) only for the rows indexed by $j \in \Omega_t$. After obtaining the candidate iterate $\hat{\boldsymbol{F}}$ above, the final step updates $\boldsymbol{F}_t$ by averaging in (14).

---

**Algorithm 1:** SHASTA-PCA

---

**Input:** Rank $k$, weights $(w_t) \in (0, 1]$, parameters $c_F, c_v > 0$, initialization parameter $\delta > 0$.
**Data:** $[\boldsymbol{y}_1, \ldots, \boldsymbol{y}_T]$, $\boldsymbol{y}_t \in \mathbb{R}^d$, group memberships $g_t \in \{1, 2, \ldots, L\}$ for all $t$, and sets of observed indices $(\Omega_1, \ldots, \Omega_T)$, where $\Omega_t \subseteq \{1, \ldots, d\}$.
**Output:** $\boldsymbol{F} \in \mathbb{R}^{d \times k}$, $\boldsymbol{v} \in \mathbb{R}_+^L$.

**1** Initialize $\boldsymbol{F}_0$ and $\boldsymbol{v}_0$ via random initialization;

**2** Initialize surrogate parameters $\overline{\boldsymbol{R}}_{t,j} = \delta \boldsymbol{I}_k$ for $\delta > 0$ and $\overline{\boldsymbol{s}}_{t,j} = \boldsymbol{0}_k$ for $j \in [d]$, $\overline{\boldsymbol{\theta}}_0 = \overline{\boldsymbol{\rho}}_0 = \boldsymbol{0}_L$ ;

**3 for** $t = 1, \ldots, T$ **do**

**4** $\quad$ Fixing $\boldsymbol{F}$ at $\boldsymbol{F}_{t-1}$,

$\qquad$ 1. Compute $\overline{\boldsymbol{\theta}}_t$ and $\overline{\boldsymbol{\rho}}_t$ via (15)-(17).

$\qquad$ 2. Compute $\boldsymbol{v}_t$ from (18).

$\quad$ Fixing $\boldsymbol{v}$ at $\boldsymbol{v}_t$,

$\qquad$ 1. Update $\overline{\boldsymbol{R}}_{t,j}$ and $\overline{\boldsymbol{s}}_{t,j}$ via (20)-(23).

$\qquad$ 2. Compute $\hat{\boldsymbol{F}}$ via (24) in parallel.

$\qquad$ 3. $\boldsymbol{F}_t = (1 - c_F)\boldsymbol{F}_{t-1} + c_F \hat{\boldsymbol{F}}$.

---

### 6.3 Algorithm and Memory/Computational Complexity

Together, these alternating updates form the Streaming HeteroscedASTic Algorithm for PCA (SHASTA-PCA), detailed in Algorithm 1.

The primary memory requirement of SHASTA-PCA is storing $d + 1$ many $k \times k$ matrices and $k$-length vectors for the $\boldsymbol{F}$ surrogate parameters and two additional $L$-length vectors for the $\boldsymbol{v}$ parameters. Thus, the dominant memory requirement of SHASTA-PCA is $\mathcal{O}(d(k^2 + k))$ memory elements throughout the runtime, which is independent of the number of data samples.

The primary sources of computational complexity arise from: i) forming $\boldsymbol{M}_t(\cdot, \cdot)$ at a cost of $\mathcal{O}(|\Omega_t|k^2 + k^3)$ flops, ii) computing $\check{\boldsymbol{z}}_t(\cdot, \cdot)$ at a cost of $\mathcal{O}(|\Omega_t|k^2)$ flops, iii) forming $\frac{1}{v_{t,g_t}} \check{\boldsymbol{z}}_t(\cdot, \cdot)\check{\boldsymbol{z}}_t(\cdot, \cdot)' + \boldsymbol{M}_t(\cdot, \cdot)$ at a cost of $\mathcal{O}(k^2)$ flops, iv) computing $\rho_{t,\ell}$ at a cost of $\mathcal{O}(|\Omega_t|k^2 + k^3)$ flops when using an efficient implementation with matrix-vector multiplications, and v) updating $\boldsymbol{F}_t$ at a cost of $\mathcal{O}(|\Omega_t|(k^3 + k^2))$ for the multiplications and inverses. In total, each iteration of SHASTA-PCA incurs $\mathcal{O}(|\Omega_t|(k^3 + k^2))$ flops.

As discussed below, PETRELS (Chi et al., 2013) uses rank-one updates to the pseudo-inverses of the matrices in (26) to avoid computing a new pseudo-inverse each iteration, but that approach does not apply in our case since the updates to $\overline{\boldsymbol{R}}_{t,j}$ in (20) are not rank-one. Still, updating $\boldsymbol{F}$ only requires inverting $|\Omega_t|$ many $k \times k$ matrices each iteration, which remains relatively inexpensive since $k \ll d$ and is often small in practice. Note that the complexity appears to be the worst for $|\Omega_t| = d$ with the implementation described above, but in this setting, since all the surrogate parameters are the same, one can verify that only a single surrogate parameter $\overline{\boldsymbol{R}}_t$ (and its inverse) is necessary. In reality, the worst-case complexity happens when $|\Omega_t| = d - 1$, i.e., when a single entry per column is missing. Identifying an approach with improved computational complexity in the highly-sampled setting remains an interesting future research direction.

#### 6.3.1 Convergence

Empirically, we observe that SHASTA-PCA converges to a stationary point as the number of samples grows. Several factors influence how fast the algorithm converges in practice. Similar to stochastic gradient descent, the choices of weights $(w_t)$ and iterate averaging parameters $c_F$ and $c_v$ affect both how fast the algorithm converges and what level of accuracy it achieves. Using larger weights tends to lead to faster convergence but only to within a larger, suboptimal local region of an optimum. Conversely, using smaller weights tends

to lead to slower progress but to a tighter region around an optimum. The amount of missing data also plays a major role. A higher percentage of missing entries generally requires more samples or more passes over the data to converge to an optimum.

Our setting differs in several key ways from the prior works discussed in §2 that establish convergence for SMM algorithms. First, our minorizers are neither Lipschitz smooth nor strongly concave (in fact, the minorizer for $v_\ell$ is nonconcave), so the theory in Mairal (2013) and Mensch et al. (2017) does not directly apply. Second, our algorithm maximizes the log-likelihood in two blocks of variables, and does so in an alternating fashion with two separate approximate minorizers $\bar{\Psi}_t^{(F)}(\boldsymbol{F})$ and $\bar{\Psi}_t^{(v)}(\boldsymbol{v})$, which is distinct from the work in Lyu et al. (2022) that alternates updates over the blocks of a single joint approximate minorizer. Notably, such as in the case of $\boldsymbol{v}$, we update an aggregation $\bar{\Psi}_t^{(v)}(\boldsymbol{v})$ of the restricted minorizers $\Psi_i(\boldsymbol{F}_{i-1}, \boldsymbol{v}; \boldsymbol{F}_{i-1}, \boldsymbol{v}_{i-1})$:

$$\bar{\Psi}_t^{(v)}(\boldsymbol{v}) = (1 - w_t)\bar{\Psi}_{t-1}^{(v)}(\boldsymbol{v}) + w_t\Psi_t(\boldsymbol{F}_{t-1}, \boldsymbol{v}; \boldsymbol{F}_{t-1}, \boldsymbol{v}_{t-1})$$

$$= \sum_{i=0}^{t} \left[ w_i \prod_{j=i+1}^{t} (1 - w_j) \right] \Psi_i(\boldsymbol{F}_{i-1}, \boldsymbol{v}; \boldsymbol{F}_{i-1}, \boldsymbol{v}_{i-1}). \tag{25}$$

The dependence of $\bar{\Psi}_t^{(v)}(\boldsymbol{v})$ on all past iterates $\{\boldsymbol{F}_i\}_{i=0}^t$ in the first argument of each $\Psi_i$ in (25) precludes using the analysis of Lyu et al. (2022) for BCD of a single approximate minorizer each iteration. Likewise, the use of this restricted minorizer in $\bar{\Psi}_t^{(v)}(\boldsymbol{v})$ precludes using the analysis of Mairal (2013) since it only minorizes the objective with respect to $\boldsymbol{v}$ with $\boldsymbol{F}$ fixed at $\boldsymbol{F}_{i-1}$, which breaks, e.g., the induction argument used in Mairal (2013, Proposition 3.3). To our knowledge, no existing work establishes convergence for this setting.

However, we conjecture that similar convergence guarantees are possible for SHASTA-PCA since both $\bar{\Psi}_t^{(F)}(\boldsymbol{v})$ and $\bar{\Psi}_t^{(v)}(\boldsymbol{v})$ retain many of the same MM properties used to analyze SMM algorithms. It is likely that a theory of convergence to a stationary point would require adjustments to our algorithm, such as the addition of constraints on $\boldsymbol{F}$ and $\boldsymbol{v}$ to provide some control over the curvature of the objective. Since the convergence results possible for non-convex problems using stochastic optimization are typically weak, we leave the proof of convergence to future work.

### 6.4 Connection to recursive least squares and HePPCAT

The SHASTA-PCA update for the factors in (24) resembles recursive least squares (RLS) algorithms like PE-TRELS (Chi et al., 2013). The objective function considered in Chi et al. (2013) is, in fact, the maximization of our complete log-likelihood in the homoscedastic setting with respect to the factors $\boldsymbol{F}$ and latent variables $\boldsymbol{z}_t$ without the $\ell_2$ penalty on $\boldsymbol{z}_t$ in (4). PETRELS first estimates the minimizer to $\boldsymbol{z}_t$ via the pseudo-inverse solution and then updates each row $\boldsymbol{f}_j$ by computing (24) using similar updates to the minorizer parameters:

$$\begin{aligned} \text{for } j \in \Omega_t: \quad & \bar{\boldsymbol{R}}_{t,j} = \lambda \bar{\boldsymbol{R}}_{t-1,j} + \hat{\boldsymbol{z}}_t \hat{\boldsymbol{z}}_t', \quad & \bar{\boldsymbol{s}}_{t,j} = \lambda \bar{\boldsymbol{s}}_{t-1,j} + y_{t,j} \hat{\boldsymbol{z}}_t, \\ \text{for } j \notin \Omega_t: \quad & \bar{\boldsymbol{R}}_{t,j} = \lambda \bar{\boldsymbol{R}}_{t-1,j}, \quad & \bar{\boldsymbol{s}}_{t,j} = \lambda \bar{\boldsymbol{s}}_{t-1,j}, \end{aligned}$$

where $\hat{\boldsymbol{z}}_t = \boldsymbol{F}_{t-1,\Omega_t}^\dagger \boldsymbol{y}_{\Omega_t}$ and $\lambda \in (0, 1)$ is a forgetting factor that exponentially downweights the importance of past data. In the stochastic MM framework, $w_t$ plays an analogous role to $\lambda$ by exponentially down-weighting surrogates constructed from historical data.

However, there are some important differences. Here, the complete data log-likelihood effectively introduces Tikhonov regularization on $\boldsymbol{z}_t$, where the Tikhonov regularization parameter is learned by estimating the noise variances. The PETRELS objective function can similarly incorporate regularization on the weights, but with a user-specified hyperparameter. It is well known that the appropriate hyperparameter in Tikhonov regularization depends on the noise variance of the data (O'Leary, 2001; Cao et al., 2021). Here, SHASTA-PCA implicitly learns this hyperparameter as part of the maximum-likelihood estimation problem for the unknown heterogeneous noise variances.

PETRELS can also be thought of as a stochastic second-order method that quadratically majorizes the function in $\boldsymbol{F}$ at each time $t$ using the pseudo-inverse solution of the weights given some estimate $\boldsymbol{F}_{t-1}$.

Our algorithm optimizes a similar quadratic majorizer in $\boldsymbol{F}$ for each $t$. While the pseudo-inverse solution for maximizing the complete data log-likelihood in (4) with respect to $\boldsymbol{z}_t$, or equivalently the conditional mean of $\boldsymbol{z}_t$, appears in the update of $\boldsymbol{F}$ through $\check{\boldsymbol{z}}_t$, the update additionally leverages the covariance of the latent variable's conditional distribution and, perhaps most importantly, an inverse weighting according to the learned noise variances that downweights noisier data samples.

SHASTA-PCA also has connections to the HePPCAT algorithm (Hong et al., 2021). Indeed, the SHASTA-PCA updates of $\boldsymbol{F}$ and $\boldsymbol{v}$ closely resemble—and can be interpreted as stochastic approximations to—HePPCAT's EM updates of $\boldsymbol{F}$ and $\boldsymbol{v}$ in Hong et al. (2021, eqn. (8)) and Hong et al. (2021, eqn. (15)), respectively. More precisely, each $\bar{\boldsymbol{s}}_{t,j}$ approximates each column of

$$\sum_{\ell=1}^{L} \frac{\check{\boldsymbol{Z}}_{t,\ell}\boldsymbol{Y}_\ell'}{v_{t,\ell}}$$

of Hong et al. (2021, eqn. (8)), where $\check{\boldsymbol{Z}}_{t,\ell} \triangleq \boldsymbol{M}_{t,\ell}\boldsymbol{F}_t'\boldsymbol{Y}_\ell$ and $\boldsymbol{M}_{t,\ell} \triangleq (\boldsymbol{F}_t'\boldsymbol{F}_t + v_{t,\ell}\boldsymbol{I}_k)^{-1}$, and each $\bar{\boldsymbol{R}}_{t,j}$ approximates the matrix

$$\sum_{\ell=1}^{L} \frac{\check{\boldsymbol{Z}}_{t,\ell}\check{\boldsymbol{Z}}_{t,\ell}'}{v_{t,\ell}} + n_\ell \boldsymbol{M}_{t,\ell} .$$

However, each of these terms in SHASTA-PCA depends on the observed data coordinate in the update of the corresponding row of $\boldsymbol{F}$. Since each row of $\boldsymbol{F}$ depends on a different $\bar{\boldsymbol{R}}_{t,j}$ for each $j \in [d]$ due to missing data, we cannot use the SVD factorization of $\boldsymbol{F}_t$ to expedite the inverse computation as in HePPCAT (Hong et al., 2021, eqn. (9)). Other minorizers for the $\boldsymbol{v}$ update that were considered in Hong et al. (2021), such as the difference of concave, quadratic solvable, and cubic solvable minorizers, may also have possible stochastic implementations. We leave these possible approaches to future work since they did not appear to result in more efficient updates.

## 7 Experimental Results

### 7.1 Incremental computation with static subspace

This section considers the task of estimating a static planted subspace from low-rank data corrupted by heterogeneous noise. We generate data according to the model in (1) with ambient dimension $d = 100$ from a rank-3 subspace with squared singular values $\boldsymbol{\gamma} = [4, 2, 1]$, drawing 500 samples with noise variance $10^{-2}$, and 2,000 samples with noise variance $10^{-1}$. We draw an orthonormal subspace basis matrix $\boldsymbol{U} \in \mathbb{R}^{100 \times 3}$ uniformly at random from the Stiefel manifold, and set the planted factor matrix to be $\boldsymbol{F} = \boldsymbol{U}\operatorname{diag}(\sqrt{\boldsymbol{\gamma}})$.

After randomly permuting the order of the data vectors, we compare SHASTA-PCA to PETRELS and the streaming PCA algorithm GROUSE (Balzano et al., 2010) (which has recently been shown to be equivalent to Oja's method (Oja, 1982) in Balzano (2022)) that estimates a subspace from rank-one gradient steps on the Grassmann manifold, with a tuned step size of 0.01.[1] SHASTA-PCA jointly learns both the factors $\boldsymbol{F}$ and noise variances $\boldsymbol{v}$ from each streaming observation. For SHASTA-PCA, we use $w_t = 1/t$ (where $t$ is the time index), $c_F = c_v = 0.1$ and initialize the parameters $\bar{\boldsymbol{R}}_t(i) = \delta\boldsymbol{I}$ with $\delta = 0.1$ for both SHASTA-PCA and PETRELS. We initialize each streaming algorithm with the same random $\boldsymbol{F}_0$, and each entry of $\boldsymbol{v}_0$ for SHASTA-PCA uniformly at random between 0 and 1. We set the forgetting parameter in PETRELS to $\lambda = 1$, corresponding to the algorithm's batch mode. As a baseline, we compare to batch algorithms for fully-observed data: HePPCAT (Hong et al., 2021) with 100 iterations, which we found to be sufficient for convergence, and homoscedastic probabilistic PCA (PPCA) (Tipping & Bishop, 1999b) on the full data. In addition, we compute PPCA on each data group individually, denoted by "G1" ("G2") in the legend of Fig. 1a corresponding to group 1 (2) with 500 (2,000) samples with noise variances $10^{-2}$ ($10^{-1}$) respectively.

The first experiment in Fig. 1a compares each algorithm in the fully observed data setting, where the streaming algorithms compute $\boldsymbol{F}$ and $\boldsymbol{v}$ incrementally using a single vector in each iteration. Given the

---

[1]All experiments were performed in Julia on a 2021 Macbook Pro with the Apple M1 Pro processor and 16 GB of memory. We reproduced and implemented all algorithms ourselves from their original source works.

planted model parameters $\boldsymbol{F}^*$ and $\boldsymbol{v}^*$ and their log-likelihood value $\mathcal{L}^* := \mathcal{L}(\boldsymbol{F}^*, \boldsymbol{v}^*)$, the left plot in Fig. 1a shows the normalized log-likelihood $\mathcal{L}(\boldsymbol{F}_t, \boldsymbol{v}_t) - \mathcal{L}^*$ with respect to the full dataset in (2) for each iteration of SHASTA-PCA compared to the batch algorithm baselines. Because GROUSE and PETRELS do not estimate the noise variances, we omit them from this plot. The right plot in Fig. 1a shows convergence of the $\boldsymbol{F}$ iterates with respect to the normalized subspace error $\frac{1}{k}\|\hat{\boldsymbol{U}}_t\hat{\boldsymbol{U}}_t' - \boldsymbol{U}\boldsymbol{U}'\|_F^2$ for the estimate $\hat{\boldsymbol{U}}_t \in \mathbb{R}^{d \times k}$ of the planted subspace $\boldsymbol{U}$; for SHASTA-PCA and PETRELS, we compute the estimated subspace $\hat{\boldsymbol{U}}_t$ by orthogonalizing $\boldsymbol{F}$. Each figure plots the mean of 50 random initializations in bold dashed traces, where their standard deviations are displayed as ribbons. The experiment in Fig. 1b then subsamples 50% of the data entries uniformly at random, inserting zeros for the missing entries for methods that require fully sampled data, and compares the same statistics across the algorithms.

As expected, when the samples are fully observed, PETRELS converges to the same log-likelihood and subspace error for each set of training data as the batch algorithms that assume homoscedastic noise, and SHASTA-PCA converges to the same log-likelihood value and subspace error as HePPCAT. Here we see the advantage of using heteroscedastic data analysis. Instead of discarding samples from either data group or combining them in a single PPCA, the heteroscedastc PPCA algorithms leverage both the "clean" samples, the additional "noisy" samples, and the noise variance estimates to produce better subspace estimates. With many missing entries (imputed with zeros), the batch algorithms' subspace estimates quickly deteriorate, as seen on the right-hand side of Fig. 1b. Of the streaming PCA algorithms for missing data, SHASTA-PCA again attains the best subspace estimate compared to GROUSE and PETRELS.

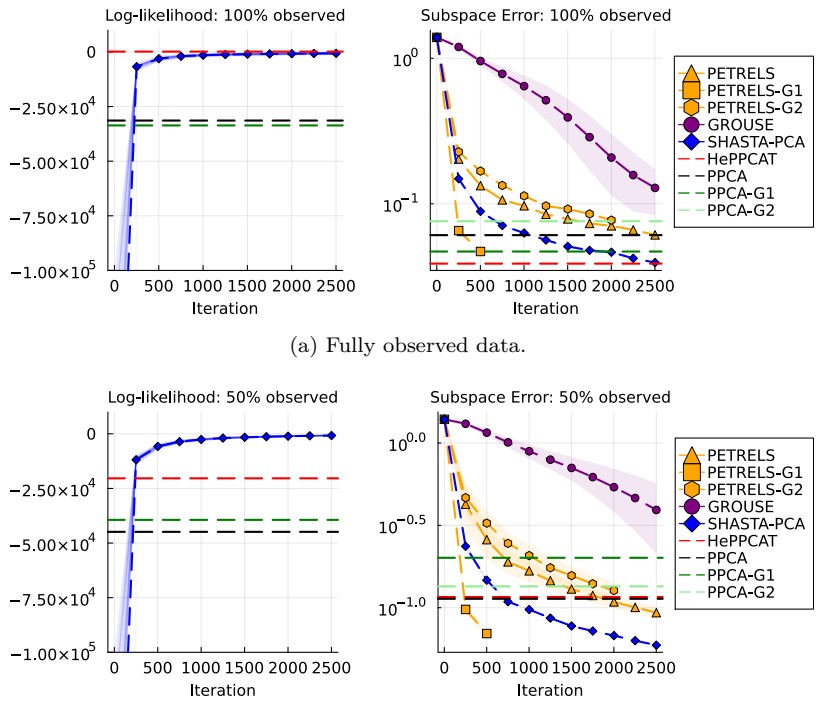

(a) Fully observed data.

(b) Data with 50% entries observed uniformly at random.

Figure 1: Incremental computation (with one pass) over batch data generated from a static subspace for $d = 100$, $n_1 = 500$, $n_2 = 2{,}000$, $v_1 = 10^{-2}$, and $v_2 = 10^{-1}$. Horizontal dashed lines show the terminal values for the batch algorithms, and the horizontal axis shows the iteration index for the online algorithms.

## 7.2 Dynamic subspace

This section studies how well SHASTA-PCA can track a time-varying subspace. We generate 20,000 streaming data samples according to the model in (1) for $L = 2$ groups with noise variances $v_1 = 10^{-4}$ and $v_2 = 10^{-2}$. We use a randomly drawn $\boldsymbol{F} = \boldsymbol{U} \operatorname{diag}(\sqrt{\boldsymbol{\gamma}})$, where $d = 100$ and $\boldsymbol{\gamma} = [4, 2, 1]$. The data

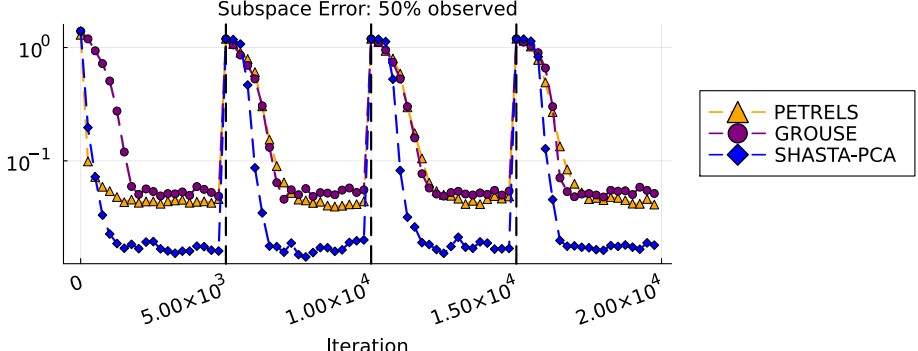

Figure 2: Dynamic tracking of rapidly shifting subspace with 50% of the entries observed uniformly at random using SHASTA-PCA versus streaming PCA algorithms that assume homoscedastic noise. Here, $d = 100$ and 20% of the data has noise variance $10^{-4}$ and 80% of the data has noise variance $10^{-2}$. Iterations refers to the number of streamed data vectors.

samples are drawn from the two groups with 20% and 80% probability, respectively. We then observe 50% of the entries selected uniformly at random. To simulate dynamic jumps of the model, we set the planted subspace $\boldsymbol{U}$ to a new random draw every 5000 samples and compare the subspace errors of the various methods with respect to the current $\boldsymbol{U}$ over time. Here, we empirically selected the constant parameters $w_t = 0.01, c_F = 0.01$, and $c_v = 0.1$ for SHASTA-PCA that do not decay with time to adaptively track the dynamics of the subspace. After hyperparameter tuning, we set the step size of GROUSE to be 0.02 and set the forgetting factor for PETRELS to be $\lambda = 0.998$. Each algorithm is initialized with the same random factors $\boldsymbol{F}_0$, and SHASTA-PCA's noise variances are initialized uniformly at random between 0 and 1. Fig. 2 shows SHASTA-PCA outperforms the streaming PCA algorithms that assume homoscedastic noise by half an order of magnitude. The results highlight how the largest noise variance dominates the streaming PCA algorithms' subspace tracking performance while SHASTA-PCA obtains more faithful estimates by accounting for the heterogeneity.

### 7.3 Dynamic noise variances

In some applications, due to temperature, age, or a change in calibration, the quality of the sensor measurements may also change with time (Jun-hua et al., 2003), thereby affecting the levels of noise in the data. To study the performance of SHASTA-PCA in these settings, we generate samples from the planted model described above where we change the noise variances over time while keeping the subspace stationary. As before, SHASTA-PCA is initialized from a random $(\boldsymbol{F}_0, \boldsymbol{v}_0)$. Figs. 3a and 3c show the estimated noise variances and the subspace error as we double the noise variance of the first group every 5,000 samples. Figs. 3b and 3d repeat the experiment but double the noise variance of the second group instead. As $v_1$ increases and the cleaner group becomes noisier, the data becomes noisier overall and also closer to homoscedastic. SHASTA-PCA's estimate of the subspace degrades and approaches the estimates obtained by PETRELS and GROUSE. On the other hand, as the noisier group gets even noisier, the quality of the PETRELS subspace estimate deteriorates over time, whereas SHASTA-PCA remains robust to the added noise by leveraging the cleaner data group. In both instances, GROUSE appears to oscillate about an optimum in a region whose size depends on the two noise variances. The variance estimates demonstrate how SHASTA-PCA can quickly adapt to changes in the noise variances; SHASTA-PCA adapted here in less than 1,000 samples.

### 7.4 Comparison to streaming robust PCA

This section experimentally compares heteroscedastic PCA and robust PCA methods. A variety of robust subspace tracking techniques exist in the literature, including for problems with missing data; see He et al. (2012); Bouwmans & Zahzah (2014); Zhan et al. (2016); Vaswani et al. (2018); Liu et al. (2020); Dung et al. (2021); Thanh et al. (2021). Many of these works model the data as a true signal lying in a low-dimensional

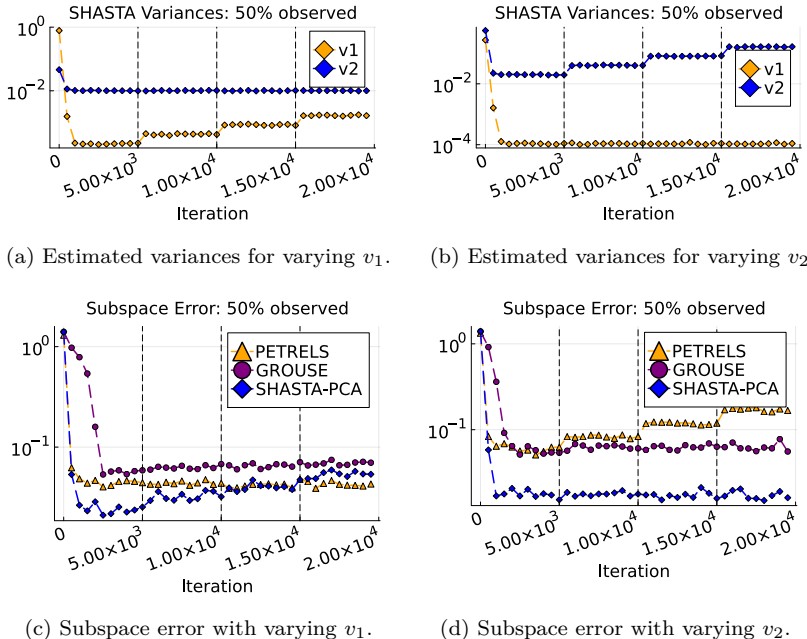

(a) Estimated variances for varying $v_1$.

(b) Estimated variances for varying $v_2$

(c) Subspace error with varying $v_1$.

(d) Subspace error with varying $v_2$.

Figure 3: Experiments with changing variances across time (iterations) for a single $\boldsymbol{F} \in \mathbb{R}^{100 \times 3}$ starting with planted noise variances $\boldsymbol{v} = [10^{-4}, 10^{-2}]$ with 50% of the entries observed uniformly at random. The vertical dashed lines indicate points at which we double one of the planted variances. The top plots show the estimated variances from SHASTA-PCA, and the bottom plots show the subspace error.

subspace, plus Gaussian noise, plus a sparse outlier vector. Given mild conditions on the incoherence and dimension of the subspace as well as the sparsity and distribution of outliers' support, robust PCA provably decomposes low-rank plus sparse data (Chandrasekaran et al., 2011).

Here, we compare SHASTA-PCA with PETRELS-ADMM (Thanh et al., 2021), a streaming robust PCA algorithm[2]. We consider subsets of data with increasingly larger differences in noise variances and varying ratios of "clean" to "noisy" samples. In one scenario, a minority of samples with a very large noise variance can be interpreted as outliers. We repeat the experiments in Section 7.1 for one random data draw with varying choices of $\boldsymbol{v}$ and missing data rates. For each experiment, we vary the sparsity hyperparameter of PETRELS-ADMM in increments of 0.1 between 0.1 and 2, choosing the hyperparameter that gave the lowest subspace error. We use a forgetting factor of $\lambda = 1$ for both PETRELS and PETRELS-ADMM, and we use the same SHASTA-PCA hyperparameters as the experiments in Section 7.1.

Although PETRELS-ADMM models for sparse entry-wise outliers, our experiments in Fig. 4 show that it achieved lower subspace estimation error than PETRELS when the two noise variances differ significantly. This suggests that the low-rank-plus-sparse PCA model improves subspace estimation even under model mismatch and when entire groups of samples are heavily contaminated with noise. Notably, Fig. 4 shows that SHASTA-PCA matched or outperformed PETRELS-ADMM in low-SNR scenarios and consistently achieved a lower subspace error when the data contained missing entries. However, SHASTA-PCA assumes and requires knowledge of the data group memberships. When this assumption is met, streaming heteroscedastic PCA offers a robust solution for data with outlier groups.

## 7.5 Computational timing experiments

Computational and/or storage considerations can inhibit the use of batch algorithms for large datasets, especially on resource constrained devices. To demonstrate the benefit of SHASTA-PCA in such settings,

---
[2]We used ChatGPT to convert the authors' original MATLAB code (`https://github.com/thanhtbt/RST`) to Julia, and we validated its outputs.

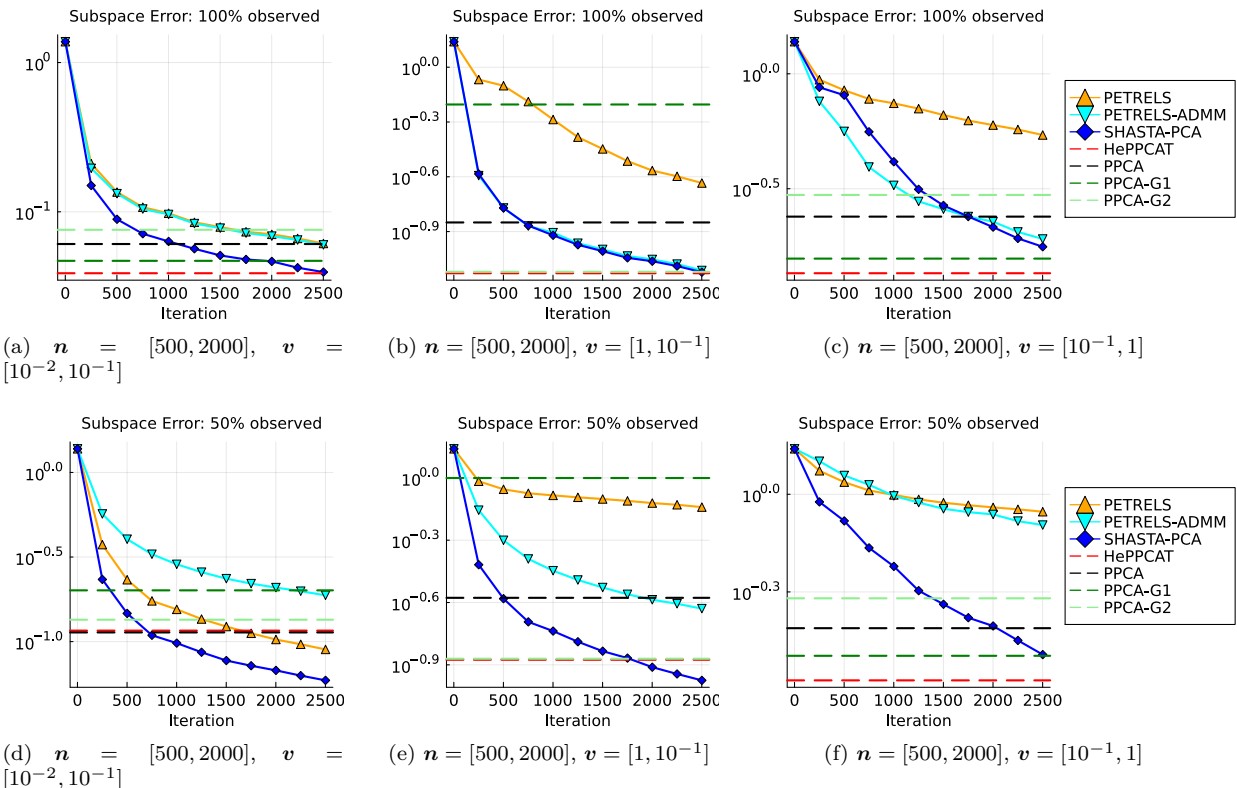

Figure 4: Experiments comparing SHASTA-PCA (this work), PETRELS (Chi et al., 2013), and PETRELS-ADMM (Thanh et al., 2021) for synthetic data.

we generate a 2GB dataset according to our model, where $d = 1,000$, $\boldsymbol{\gamma} = [4, 2, 1]$, $\boldsymbol{n} = [50,000, \ 200,000]$, $\boldsymbol{v} = [0.1, 1]$, and we observed only 20% of the entries uniformly at random. For this experiment, we set $w_t = 0.01/\sqrt{t}$, $c_F = 0.01$, and $c_v = 0.1$ for SHASTA's hyperparameters, and pass over the entire data once. Fig. 5 compares the convergence in log-likelihood values and subspace errors by elapsed wallclock time for SHASTA and the batch algorithm in §5, where both algorithms are randomly initialized from the same random starting iterate $(\boldsymbol{F}_0, \boldsymbol{v}_0)$. SHASTA-PCA rapidly obtains a good estimate of the model, using only roughly 60% of the time that it took the batch method, all while using only 0.0048% of the memory per iteration.

## 7.6 Real data from astronomy

We illustrate SHASTA-PCA on real astronomy data from the Sloan Digital Sky Survey (SDSS) Data Release 16 (Ahumada et al., 2020) using the associated DR16Q quasar catalog (Lyke et al., 2020). In particular, we consider the subset that was considered in Hong et al. (2023, Section 8); see Hong et al. (2023, Supplementary Material SM5) for details about the subset selected and the preprocessing performed. The dataset contains $n = 10{,}459$ quasar spectra, where each spectrum is a vector of $d = 281$ flux measurements across wavelengths and the data come with associated noise variances.

Ordering the samples from the smallest to the largest noise variance estimates, we obtained a "ground-truth" signal subspace by taking the left $k = 5$ singular vectors of the data matrix for the first 2,000 samples with the smallest noise variance estimates. We then formed a training dataset with two groups: first, we collected samples starting from sample index 6,500 to the last index where the noise variance estimate is less than or equal to 1 (7,347); second, we collected training data beginning at the first index where the noise variance estimate is greater than or equal to 2 (8,839) up to the sample index 10,449, excluding the last 10 samples that are grossly corrupted. The resulting training dataset had $n_1 = 848$ and $n_2 = 1{,}611$ samples for the

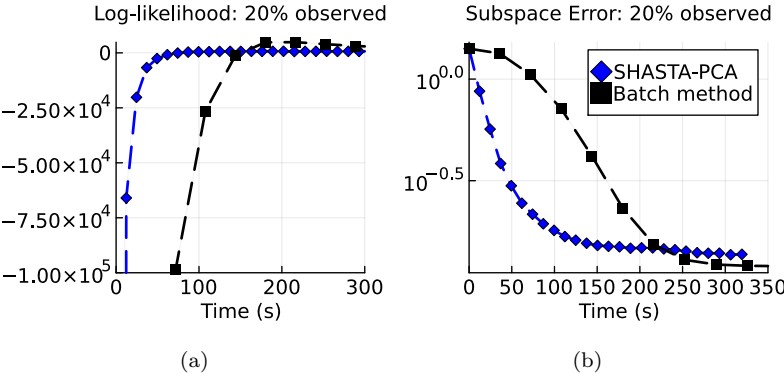

(a)                                                      (b)

Figure 5: Log-likelihood values and subspace errors versus elapsed wall clock time for one run of SHASTA-PCA versus the batch method in §5 on 2GB of synthetic data: $d = 1{,}000$, $k = 3$, $\boldsymbol{n} = [50{,}000, \ 200{,}000]$, and $\boldsymbol{v} = [0.1, 1]$. Both algorithms used the same random initialization. Markers for SHASTA-PCA are for every 10,000 vector samples, and markers for the batch method are for each algorithm iteration.

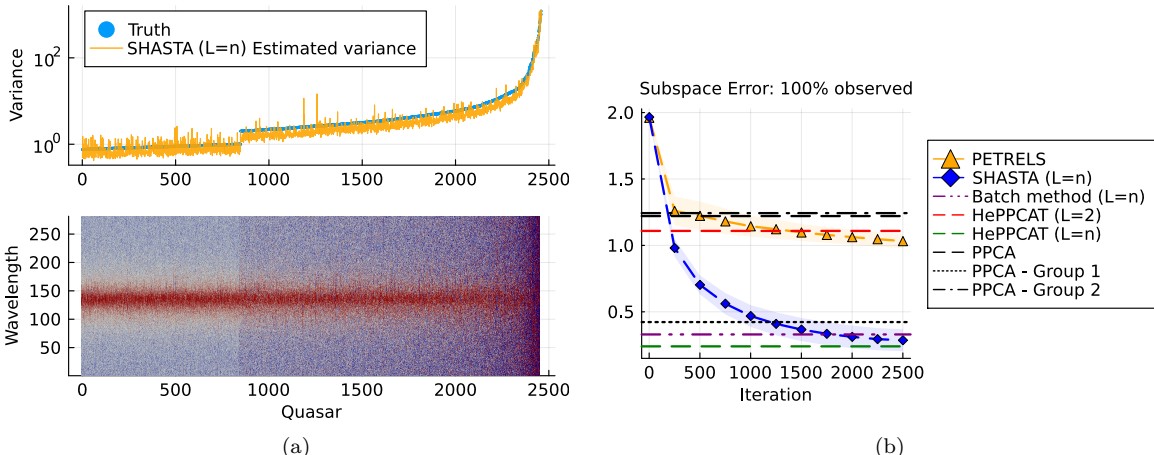

(a)                                                      (b)

Figure 6: (a) Visualization of quasar data with true associated variances of each sample plotted by quasar. The estimated variances from a SHASTA-PCA $L = n$ model streaming over the samples (fully sampled, in randomized order) closely matched the true variances. (b) For fully sampled data, the subspace error of SHASTA-PCA converged to the error of HePPCAT's estimated subspace for a $L = n$ model. We repeated the experiment 50 times, each with a different random initialization and order of the samples.

two groups, respectively, and had strong noise heterogeneity across the samples, where the second group was much noisier than the first. Fig. 6a shows the training dataset and the associated noise variance estimates for each sample.

Although we formed the data by combining two groups of consecutive samples, the data actually contain $L = n$ groups since each spectrum has its own noise variance. This setting allows for a heuristic computational simplification in SHASTA-PCA. Namely, we adapt the $L = 1$ model for a single variance $v$ and use separate weights for the SMM updates of $\boldsymbol{F}$ and $v$, where $w_t^{(F)} = 0.001$, $c_F = 0.1$, and $w_t^{(v)} = c_v = 1 \ \forall t$. The variance update is then equivalent to maximizing $\Psi_t(\boldsymbol{F}_{t-1}, v; \boldsymbol{F}_{t-1}, v_{t-1})$, i.e., the minorizer centered at the previous variance estimate with no memory of previous minorizers. The number of variance EM updates per data vector may be increased beyond just a single update, but in practice we observed little additional benefit. Fig. 6 shows how SHASTA-PCA adaptively learns the unknown variances for each new sample and converges to the same level of subspace error as the batch HePPCAT $L = n$ model.

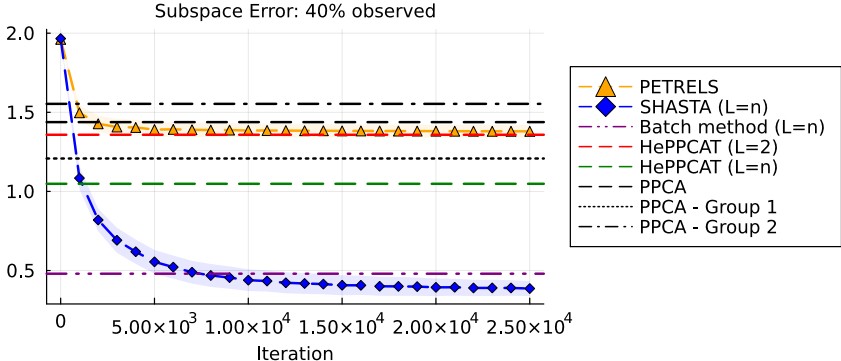

Figure 7: Subspace error for quasar data with 40% of the entries observed uniformly at random. SHASTA-PCA streams over the full dataset 10 times, with the sample entries missing, where the order of the samples was randomized on each pass. Each experiment was initialized with factors chosen uniformly at random.

In many modern large datasets, entries may be missing in significant quantities due to sensor failure or time and memory constraints that preclude acquiring complete measurements. Indeed, the experiment designer may only wish to measure a "sketch" of the full data to save time and resources and learn the underlying signal subspace from limited observations using an algorithm like SHASTA-PCA. To study this case, we randomly obscure 60% of the entries uniformly at random and perform 10 passes over the data, randomizing the order of the samples each time. We use the same choice of weights described above to estimate a single variance for every new sample. We initialize with random $\boldsymbol{F}_0$ and $v_0$ using the zero-padded data. As Fig. 7 shows, SHASTA-PCA has better subspace estimates in this limited sampling setting than the state-of-the-art baseline methods with zero-filled missing entries and/or homoscedastic noise assumptions. Interestingly, the SHASTA-PCA subspace estimate is even better than the batch method in §5 for the $L = n$ model.

## 8 Conclusion & Future Work

This paper proposes a new streaming PCA algorithm (SHASTA-PCA) that is robust to *both* missing data and heteroscedastic noise across samples. SHASTA-PCA requires only a modest amount of memory that is independent of the number of samples and has efficient updates that can scale to large datasets. The results showed significant improvements over state-of-the-art streaming PCA algorithms in tracking nonstationary subspaces under heteroscedastic noise and significant improvement over a batch algorithm in speed.

There are many future directions building on this work. An interesting line of future work is to establish convergence guarantees for SHASTA-PCA, particularly since our optimization approach is unique among other works using stochastic MM. Second, while each update of a row in $\boldsymbol{F}$ is relatively cheap, it still requires inverting $|\Omega_t|$-many $k \times k$ matrices per data vector, which is particularly wasteful when $|\Omega_t| = d$ since this is equivalent to the fully-sampled log-likelihood, which only requires inverting a single $k \times k$ matrix to update $\boldsymbol{F}$. It may be possible to find other surrogate functions that would avoid this large number of small inverses in each iteration. Finally, although SHASTA-PCA enjoys lightweight computations in each iteration, achieving rapid convergence can depend on carefully tuning the weights $w_t$ and the parameters $c_F$ and $c_v$. Adaptively selecting these parameters with stochastic MM techniques and developing theory to guide the selection of these parameters remain open problems.

## 9 Acknowledgements

The authors acknowledge and thank Hanbaek Lyu for his helpful discussions on convergence analysis for stochastic MM algorithms.

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

## A  Notation table

Table 1 summarizes notation in the paper.

## B  Minorizer derivation

We give a short proof for the expectation step to derive the $i$th minorizer in (7). For notational simplicity, we drop the indexing on $i$ and give the proof for the case with fully-sampled data, without loss of generality. Let

$$\boldsymbol{x} \triangleq \begin{bmatrix} \boldsymbol{z} \\ \boldsymbol{y} \end{bmatrix} = \begin{bmatrix} \boldsymbol{I} \\ \boldsymbol{F} \end{bmatrix} \boldsymbol{z} + \begin{bmatrix} \boldsymbol{0} \\ \boldsymbol{n} \end{bmatrix}, \tag{26}$$

where $\boldsymbol{z} \sim \mathcal{N}(\boldsymbol{0}, \boldsymbol{I})$ and $\boldsymbol{n} \sim \mathcal{N}(\boldsymbol{0}, v\boldsymbol{I})$. Then $\boldsymbol{x} \sim \mathcal{N}\left(\boldsymbol{0}, \begin{bmatrix} \boldsymbol{I} & \boldsymbol{F}' \\ \boldsymbol{F} & v\boldsymbol{I} + \boldsymbol{F}\boldsymbol{F}' \end{bmatrix}\right)$. Using conditional expectations of multivariate Gaussian random variables,

$$\{\boldsymbol{z}|\boldsymbol{y} = \boldsymbol{a}\} \sim \mathcal{N}(\bar{\boldsymbol{\mu}}, \bar{\boldsymbol{\Sigma}}), \tag{27}$$

where

$$\bar{\boldsymbol{\mu}} = \boldsymbol{F}'(v\boldsymbol{I} + \boldsymbol{F}\boldsymbol{F}')^{-1}\boldsymbol{a} = (v\boldsymbol{I} + \boldsymbol{F}'\boldsymbol{F})^{-1}\boldsymbol{F}'\boldsymbol{a} \triangleq \boldsymbol{M}(\boldsymbol{F}, v)\boldsymbol{F}'\boldsymbol{a} \tag{28}$$

$$\bar{\boldsymbol{\Sigma}} = \boldsymbol{I} - \boldsymbol{F}'(v\boldsymbol{I} + \boldsymbol{F}\boldsymbol{F}')\boldsymbol{F} = v(v\boldsymbol{I} + \boldsymbol{F}'\boldsymbol{F})^{-1} \triangleq v\boldsymbol{M}(\boldsymbol{F}, v) \tag{29}$$

The two righthand equalities follow from applying the Sherman-Woodbury-Morrison formula, i.e., the matrix inversion lemma (Boyd & Vandenberghe, 2004, Section C.4.3). Finally, expanding the square quadratic terms of the complete log-likelihood in (4), taking the expectation with respect to $\boldsymbol{z}|\boldsymbol{y}$, plugging in the expressions for $\bar{\boldsymbol{\mu}}$ and $\bar{\boldsymbol{\Sigma}}$ above, and dropping terms constant with respect to $\boldsymbol{F}$ and $v$ yields the minorizer in (7).

Table 1: Summary of notation.

| Notation | Description |
|---|---|
| $\boldsymbol{y}_i \in \mathbb{R}^d$ | $i$th data vector |
| $\boldsymbol{F} \in \mathbb{R}^{d \times k}$ | factor matrix ($k \ll d$) |
| $\boldsymbol{z}_i \in \mathbb{R}^k$ | $i$th coefficients distributed $\mathcal{N}(0, \boldsymbol{I})$ |
| $\boldsymbol{\epsilon}_i$ | $i$th noise vector distributed $\mathcal{N}(0, v_{g_i}\boldsymbol{I})$ |
| $v_1, \ldots, v_L$ | noise variances of the $L$ data groups |
| $\boldsymbol{v} \in \mathbb{R}^L$ | vector of noise variances |
| $g_i \in \{1, \ldots, L\}$ | group index of the $i$th data vector |
| $\Omega_i \subseteq \{1, \ldots, d\}$ | set of entries observed for the $i$th sample |
| $\Omega$ | observed indices over the batch data |
| $\boldsymbol{y}_{\Omega_i} \in \mathbb{R}^{|\Omega_i|}$ | restriction of $\boldsymbol{y}_i$ to the rows defined by $\Omega_i$ |
| $\boldsymbol{F}_{\Omega_i} \in \mathbb{R}^{|\Omega_i| \times k}$ | restriction of $\boldsymbol{F}$ to the rows defined by $\Omega_i$ |
| $\boldsymbol{Y}_{\Omega}$ | partially observed dataset |
| $\mathcal{L}(\boldsymbol{Y}_{\Omega}; \boldsymbol{F}, \boldsymbol{v})$ | log-likelihood over $\boldsymbol{Y}_{\Omega}$ with respect to $\boldsymbol{F}$ and $\boldsymbol{v}$ |
| $\mathcal{L}_i(\boldsymbol{y}_{\Omega_i}; \boldsymbol{F}, \boldsymbol{v})$ | log-likelihood of the $i$th sample |
| $\mathcal{L}_i^c(\boldsymbol{F}, \boldsymbol{v})$ | complete log-likelihood of the $i$th sample |
| $\check{\boldsymbol{z}}_i(\boldsymbol{F}, \boldsymbol{v})$ | function defined in (5) for intermediate calculations |
| $\boldsymbol{M}_i(\boldsymbol{F}, \boldsymbol{v})$ | function defined in (6) for intermediate calculations |
| $\Psi_i(\boldsymbol{F}, \boldsymbol{v}; \widetilde{\boldsymbol{F}}, \tilde{\boldsymbol{v}})$ | minorizer of the $i$th log-likelihood at point $(\tilde{\boldsymbol{F}}, \tilde{\boldsymbol{v}})$ |
| $\Psi(\boldsymbol{F}, \boldsymbol{v}; \widetilde{\boldsymbol{F}}, \tilde{\boldsymbol{v}})$ | minorizer over the batch data |
| $\bar{\Psi}_t^{(v)}(\boldsymbol{v})$ | $t$th approximate minorizer with respect to $\boldsymbol{v}$ in the streaming algorithm |
| $\bar{\Psi}_t^{(F)}(\boldsymbol{F})$ | $t$th approximate minorizer with respect to $\boldsymbol{F}$ in the streaming algorithm |
| $\theta_{t,\ell}^{\mathrm{batch}}$ | total count of observed entries in the $\ell$th group |
| $\bar{\boldsymbol{\theta}}_t$ | term computed from the count of observed entries in the streaming algorithm |
| $\rho_{t,\ell}^{\mathrm{batch}}$ | data-dependent term computed in the batch algorithm's $\boldsymbol{v}$ update |
| $\bar{\rho}_t$ | data-dependent term computed in the streaming algorithm's $\boldsymbol{v}$ update |
| $\boldsymbol{R}_{t,j}^{\mathrm{batch}}$ | data-dependent $k \times k$ matrix computed in the batch algorithm's $\boldsymbol{F}$ update |
| $\bar{\boldsymbol{R}}_{t,j}$ | data-dependent $k \times k$ matrix computed in the streaming algorithm's $\boldsymbol{F}$ update |
| $\boldsymbol{s}_{t,j}^{\mathrm{batch}}$ | data-dependent $k$-length vector computed in the batch algorithm's $\boldsymbol{F}$ update |
| $\bar{\boldsymbol{s}}_{t,j}$ | data-dependent $k$-length vector computed in the streaming algorithm's $\boldsymbol{F}$ update |
| $\{w_t\}_{t \geq 0} \in (0, 1)$ | sequence of weights in the streaming algorithm |
| $c_F$ and $c_v$ | positive scalars in the streaming algorithm for updating $\boldsymbol{F}$ and $\boldsymbol{v}$. |

## C Hyperparameter sensitivity tests

Figures 8 and 9 summarize the hyperparameter sweep experiments. For the experiments in Section 7.1, we varied the constant $c_w$ in the numerator for decaying weights $w_t = c_w/t$, where $t$ is the algorithm iteration. We plotted the log-likelihoods and subspace errors in Fig. 8, showing the algorithm's performance is largely stable across many choices of hyperparameters. Similarly, for the dynamic subspace tracking experiments in Section 7.2, we plotted the subspace error of the final estimate for $\boldsymbol{F}$ across varying values of a constant weight $w = w_t$ for all $t$ and the constants $c_F = c_v$.

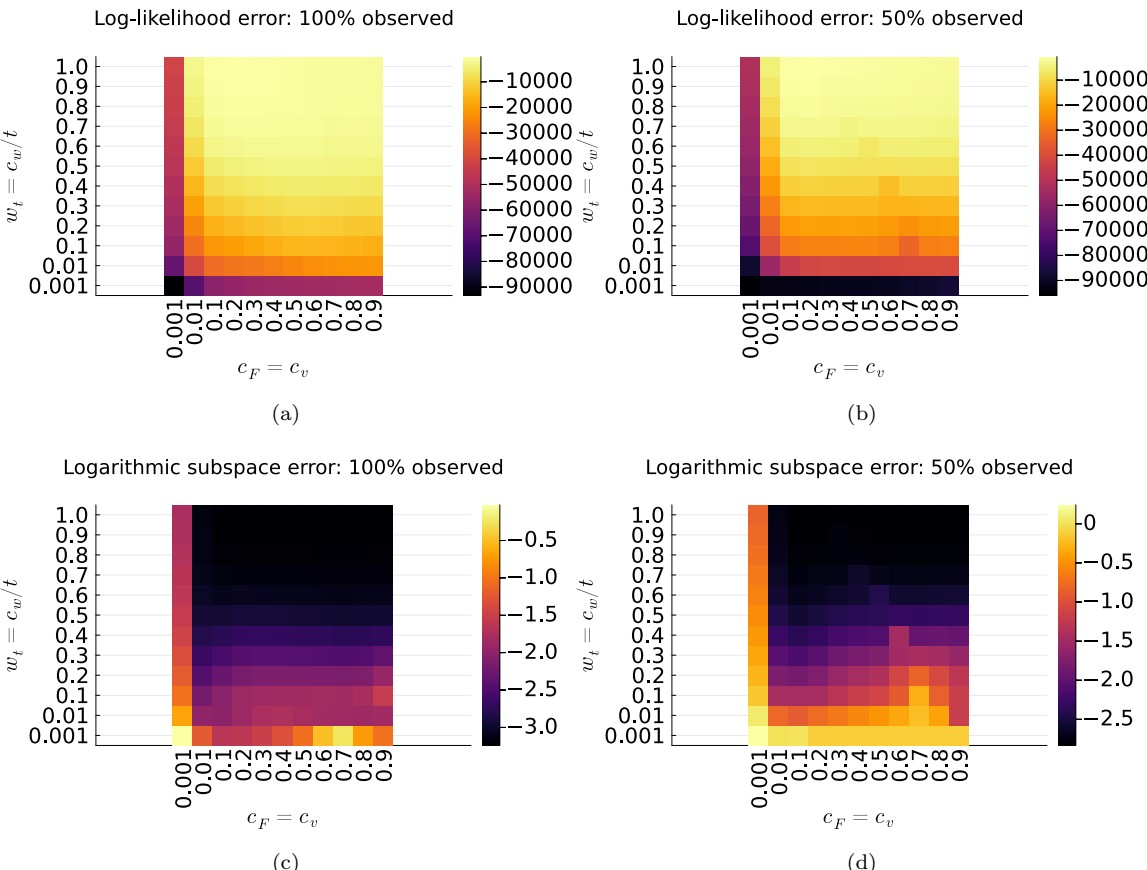

Figure 8: Hyperparameter sweep for experiments in Section 7.1.

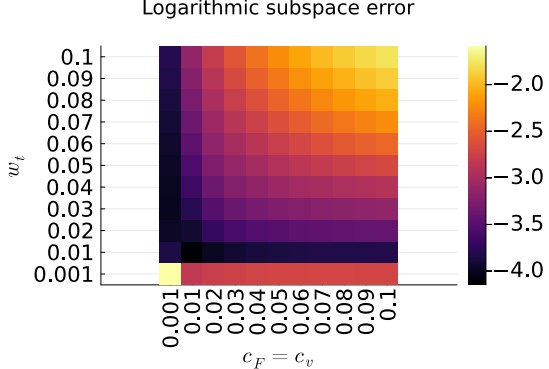

Figure 9: Hyperparameter sweep for experiments in Section 7.2.

