# OpenReview forum: "Streaming Heteroscedastic Probabilistic PCA with Missing Data"
_TMLR — Accepted by TMLR_

### Review · Reviewer_yPyN · 2025-05-14

**Summary Of Contributions:**

This paper proposes SHASTA-PCA, a streaming algorithm for probabilistic PCA that can handle both heteroscedastic noise across samples and missing data. Existing PCA methods either assume homoscedastic noise or require access to all data (full batch). Thus, a streaming variant of PCA for streaming data (with heteroscedastic noise and missing entries) is a natural extension.

**Audience:**

Yes

**Claims And Evidence:**

Yes

**Requested Changes:**

### **Major**
- The paper should be heavily revised. It could be shortened (by adding some appendices), and remove repeating sentences. Also, the author use the same notation for different quantities which is confusing to read. E.g., the authors could clearly state that $t$ is time, $i$ is the iteration, and k,l,s is the index for their sums. The authors have eq. numbers on all eqs.! but rarely refer to them...
- Add theoretical convergence analysis - similar results as in the literature - would improve the paper significantly.
- Code release or pseudo-code: Include a more detailed algorithmic description or commit to releasing code for reproducibility.
- Add notation table which summarize key symbols used throughout would improve readability.

**Strengths And Weaknesses:**

### **Strengths**
- The authors convincingly argue that real-world streaming data often contain both missing values and heteroscedastic noise, and no existing PCA method addresses both.
- The proposed method builds on stochastic MM techniques and adapts them to heteroscedastic noise and missing data.
- Experiments demonstrate good performance on both synthetic and real-world data.

### **Weaknesses**
- The paper lacks theoretical convergence results; but the authors discuss this and highlight the challenges.
-  SHASTA-PCA depends on the tuning of hyperparameters. Coming with a more adaptive or self-tuning strategy could be beneficial, in particular, in the setting of streaming data.
- The algorithm, while efficient, is relatively intricate and may be hard to reimplement without detailed code or pseudo-code beyond what is provided in the paper.

---

> ### Author Response · Authors · 2025-07-09
> **Response to Reviewer yPyN (Part I)**
>
> The authors thank the reviewer for your helpful comments. We have revised the paper to address all of your suggested changes; please see all changes in the paper in blue text. Below we respond to your comments line-by-line:
>
> 1. The paper should be heavily revised. It could be shortened (by adding some appendices), and remove repeating sentences. Also, the author use the same notation for different quantities which is confusing to read. E.g., the authors could clearly state that $t$ is time, $i$ is the iteration, and k,l,s is the index for their sums. The authors have eq. numbers on all eqs.! but rarely refer to them...
> > Thanks for these suggestions. We went through and were more thoughtful about notation; for example, we tried to keep variables with tilde (e.g. $\tilde{a}$) to refer to quantities associated with the ``current point'' of an iterative algorithm, and variables with bar (e.g. $\bar{a}$) to refer to quantities averaged over iterations.
>     We have clarified and highlighted in blue text in Sections 5 and 6 of the revised paper that $i$ refers to index of the sample, $t$ refers to algorithm iteration of the batch algorithm, and $t = i$ is the sample and SMM iteration in the streaming algorithm.
>     Equation numbers without in-text references have been omitted. Repeating sentences describing already defined shorthand notation have been omitted. We welcome any further suggestions.
>
> 2. Add theoretical convergence analysis - similar results as in the literature - would improve the paper significantly.
> >Thank you for raising this point. We agree that it would be wonderful to provide convergence guarantees for SHASTA-PCA, similar to those that exist for other stochastic majorize-minimize algorithms. Indeed, we previously searched through the existing literature for relevant results, but did not find any that could be straightforwardly applied to SHASTA-PCA. In each case, analyzing SHASTA-PCA would involve making non-trivial extensions to the existing convergence analysis for the reasons discussed in Section 6.3.1 of the paper.
> > \
> > \
> >A major source of difficulty is that SHASTA-PCA alternates between stochastic majorize-minimize updates for multiple blocks of variables (i.e., for the variance vector $v$ then for the factor matrix $F$) where each block has its own set of time-averaged minorizers ($\bar{\Psi}_{t}^{(v)}$ for $v$ and $\bar{\Psi}_t^{(F)}$ for $F$). This does not fit, e.g., within the setting of Mairal (2013) or Lyu (2024), both of which involve a single (joint) majorizer rather than individual (block-specific) majorizers.
> > \
> > \
> > For this revision, we revisited the existing analyses and worked on modifying them to obtain guarantees for SHASTA-PCA. We focused on Proposition 3.3 in Mairal(2013), which seemed the most promising.  Notably, this involved considering a slightly modified version of SHASTA-PCA in which we constrained $F$ and $v$ as follows so that the domain would have the compactness crucially needed for the arguments in Mairal(2013):
> $$
> F \in \\{F \in \mathbb{R}^{d \times k} : \left\lVert F \right\rVert_2 \leq B \\} := \Theta_F
> $$
> and
> $$
> v \in \\{v \in \mathbb{R}^L : \epsilon^2 \leq v_1, ..., v_L \leq B^2 \\} := \Theta_v,
> $$
> where $\epsilon, B > 0$ are constants. Moreover, we restricted ourselves to the setting of fully-observed data drawn i.i.d. from some compactly supported distribution, and only considered the case of SHASTA-PCA with hyperparameters $c_F = c_v = 1$.
>     However, even with these changes, we were not able to fully extend the analysis in Mairal(2013).
>     We did find ways to extend a number of the steps in the proof of Proposition 3.3 from Mairal(2013), but the primary difficulty was again due to the block-specific surrogates used in SHASTA-PCA.
>     Specifically, equation (11) of Mairal(2013) uses the fact that the time-averaged streaming objective ($\bar{f}_n$ in their notation) is bounded by the time-averaged surrogate ($\bar{g}_n$ in their notation). This holds straightforwardly in their setting by induction since each surrogate ($g_n$ in their notation) bounds the corresponding objective ($f_n$ in their notation) globally. However, in our setting, the surrogates only bound the objective along one block of variables where the other block of variables is fixed at the previous iterate. This breaks the inductive step and prevents the approach taken in Mairal(2013) from being extended to our setting. We have added a brief description of these challenges to Section 6.3.1 of the paper.
> > \
> > \
> > Given these fundamental challenges, we feel that while establishing convergence guarantees for SHASTA-PCA remains important (and we welcome any pointers to relevant references!), doing so is beyond the scope of the present paper and is best left for future work.

---

> ### Author Response · Authors · 2025-07-09
> **Response to Reviewer yPyN (Part II)**
>
> 3. Code release or pseudo-code: Include a more detailed algorithmic description or commit to releasing code for reproducibility.
> > We added pseudo-code in Algorithm 1 of the paper, and we will add a link to our Github repository with the code made publicly available once the anonymous review has concluded.
>
> 4. Add notation table which summarize key symbols used throughout would improve readability.
> > Thank you for this suggestion. Since we need to define many notation in our paper, we added a notation table to the appendix summarizing all the key symbols used.

---

### Review · Reviewer_mM8x · 2025-05-27

**Summary Of Contributions:**

In this submission, the authors introduce **SHASTA-PCA**, a novel streaming algorithm for probabilistic principal component analysis that simultaneously handles heteroscedastic noise, missing data, and non-stationary observations. At each time step $t$, one observes a partially‐observed vector $x_t\in\mathbb{R}^d$ with index set $\Omega_t\subseteq[d]$ and noise $x_t = F v_t + \varepsilon_t,\quad \varepsilon_t\sim\mathcal{N}\bigl(0,\mathrm{diag}(\sigma_t^2)\bigr),$ where $F\in\mathbb{R}^{d\times k}$ is the low‐rank factor and $\sigma_t^2\in\mathbb{R}^d$ encodes sample‐wise variances. The core technical innovation is a stochastic majorize–minimize (SMM) scheme that yields closed‐form, per‐sample updates for both $U$ and $\sigma_t^2$, avoiding expensive batch EM steps. These updates incur only $O(dk^2)$ memory and $O(|\Omega_t|k^3)$ computation per iteration.

The paper provides a a theoretical complexity analysis of per-step updates, discusses empirical convergence under mild conditions, and validates performance on synthetic data as well as real SDSS DR16 quasar spectra, demonstrating superior accuracy and substantial runtime gains over GROUSE, PETRELS, and batch heteroscedastic PCA (HePPCAT).

**Audience:**

Yes

**Claims And Evidence:**

Yes

**Requested Changes:**

I believe the paper makes a good contribution towards combining the settings of heteroscedastic noise, missing data and streaming PCA. Please see the weaknesses section above for proposed adjustments. However, they are not critical to securing my recommendation for acceptance but would strengthen the paper in my opinion.

**Strengths And Weaknesses:**

### Strengths
- Demonstrates clear improvements in subspace recovery accuracy and runtime compared to GROUSE, PETRELS, and batch heteroscedastic PCA (HePPCAT) on both synthetic and SDSS DR16 datasets.
- Successfully recovers the underlying low-rank subspace even with up to 50 % randomly missing entries per sample.
- Requires only $O(dk^2)$ memory and $O(|\Omega_t|k^3)$ compute per sample, independent of the total stream length.
- Integrates heteroscedastic noise modeling and online updates in a single end-to-end algorithm, generalizing PETRELS (streaming + missing data) and HePPCAT (batch + heteroscedastic noise).

### Weaknesses
-  Currently, the paper does not include a formal proof or sketch of conditions under which the stochastic majorize–minimize updates converge to a stationary point, even under a warm start as is typically the case for EM algorithms. Even a concise convergence guarantee or proven stability conditions would enhance confidence in the algorithm’s reliability.

- The submission does not compare against in detail against state-of-the-art adversarial/outlier-robust PCA methods such as:
  - Ilias Diakonikolas *et al.*, “Nearly-Linear Time and Streaming Algorithms for Outlier-Robust PCA” (ICML 2023): develops a nearly-linear time algorithm with near-optimal error guarantees for robust PCA and a single-pass streaming variant using nearly-linear space [\[1\]](https://arxiv.org/abs/2305.02544).
  - Eric Price and Zhiyang Xun, “Spectral Guarantees for Adversarial Streaming PCA” (FOCS 2024): provides the first adversarial streaming PCA algorithm with spectral error bounds [\[2\]](https://arxiv.org/abs/2408.10332).
  Including either an empirical comparison or a discussion of how SHASTA-PCA complements or outperforms these approaches would clarify its place in the robust PCA landscape.

- The roles of the step-size weights $w_t$ and regularization constants $(c_F, c_v)$ are explained, but practical tuning advice or an ablation study is not provided. A brief sensitivity analysis would help practitioners reproduce and adapt the method to different datasets.

---

> ### Author Response · Authors · 2025-07-09
> **Response to Reviewer mM8x (Part I)**
>
> The authors thank the reviewer for your helpful comments. We have revised the paper to address all of your suggested changes; please see all changes in the paper in blue text. Below we respond to your comments line-by-line:
>
> 1. Currently, the paper does not include a formal proof or sketch of conditions under which the stochastic majorize–minimize updates converge to a stationary point, even under a warm start as is typically the case for EM algorithms. Even a concise convergence guarantee or proven stability conditions would enhance confidence in the algorithm’s reliability.
>
> > Thank you for raising this point. We agree that it would be wonderful to provide convergence guarantees for SHASTA-PCA, similar to those that exist for other stochastic majorize-minimize algorithms. Indeed, we previously searched through the existing literature for relevant results, but did not find any that could be straightforwardly applied to SHASTA-PCA. In each case, analyzing SHASTA-PCA would involve making non-trivial extensions to the existing convergence analysis for the reasons discussed in Section 6.3.1 of the paper.
> > \
> > \
> > A major source of difficulty is that SHASTA-PCA alternates between stochastic majorize-minimize updates for multiple blocks of variables (i.e., for the variance vector $v$ then for the factor matrix $F$) where each block has its own set of time-averaged minorizers ($\bar{\Psi}_{t}^{(v)}$ for $v$ and $\bar{\Psi}_t^{(F)}$ for $F$). This does not fit, e.g., within the setting of Mairal (2013) or Lyu (2024), both of which involve a single (joint) majorizer rather than individual (block-specific) majorizers.
> > \
> > \
> > For this revision, we revisited the existing analyses and worked on modifying them to obtain guarantees for SHASTA-PCA. We focused on Proposition 3.3 in Mairal(2013), which seemed the most promising. Notably, this involved considering a slightly modified version of SHASTA-PCA in which we constrained $F$ and $v$ as follows so that the domain would have the compactness crucially needed for the arguments in Mairal(2013):
> $$
> F \in \\{ F \in \mathbb{R}^{d \times k} : \lVert F \rVert_2 \leq B \\} := \Theta_F
> $$
> and
> $$
> v \in \\{ v \in \mathbb{R}^{L} :  \epsilon^2 \leq v_1 , ... , v_L \leq B^2\\} := \Theta_v,
> $$
> where $\epsilon, B > 0$ are constants. Moreover, we restricted ourselves to the setting of fully-observed data drawn i.i.d. from some compactly supported distribution, and only considered the case of SHASTA-PCA with hyperparameters $c_F = c_v = 1$. However, even with these changes, we were not able to fully extend the analysis in Mairal(2013).
>     We did find ways to extend a number of the steps in the proof of Proposition 3.3 from Mairal(2013), but the primary difficulty was again due to the block-specific surrogates used in SHASTA-PCA.
>     Specifically, equation (11) of Mairal(2013) uses the fact that the time-averaged streaming objective ($\bar{f}_n$ in their notation) is bounded by the time-averaged surrogate ($\bar{g}_n$ in their notation). This holds straightforwardly in their setting by induction since each surrogate ($g_n$ in their notation) bounds the corresponding objective ($f_n$ in their notation) globally. However, in our setting, the surrogates only bound the objective along one block of variables where the other block of variables is fixed at the previous iterate. This breaks the inductive step and prevents the approach taken in Mairal(2013) from being extended to our setting. We have added a brief description of these challenges to Section 6.3.1 of the paper.
> > \
> > \
> >Given these fundamental challenges, we feel that while establishing convergence guarantees for SHASTA-PCA remains important (and we welcome any pointers to relevant references!), doing so is beyond the scope of the present paper and is best left for future work.

---

> > ### Comment · Reviewer_mM8x · 2025-07-17
> >
> > I appreciate the authors' detailed response and attempt at explaining the challenges in theoretical analysis. This answers my question about formal theoretical convergence guarantees.

---

> ### Author Response · Authors · 2025-07-09
> **Response to Reviewer mM8x (Part II)**
>
> 2. The submission does not compare against in detail against state-of-the-art adversarial/outlier-robust PCA methods such as:
> * Ilias Diakonikolas et al., “Nearly-Linear Time and Streaming Algorithms for Outlier-Robust PCA” (ICML 2023): develops a nearly-linear time algorithm with near-optimal error guarantees for robust PCA and a single-pass streaming variant using nearly-linear space [1].
>
> * Eric Price and Zhiyang Xun, “Spectral Guarantees for Adversarial Streaming PCA” (FOCS 2024): provides the first adversarial streaming PCA algorithm with spectral error bounds [2].
>
> Including either an empirical comparison or a discussion of how SHASTA-PCA complements or outperforms these approaches would clarify its place in the robust PCA landscape.
>
> > We appreciate this request, as the relationship of heteroscedastic PCA to robust PCA has not been explored and is of great interest. We added a comparison to PETRELS-ADMM algorithm for outlier robust PCA, and demonstrated interesting results
> that are found in Figure 4 of the revised paper. Our new experiments show that SHASTA-PCA matches or outperforms PETRELS-ADMM in low-SNR scenarios and consistently achieves a lower subspace error when the data contain missing entries.
> > \
> > \
> > We want to note that Diakonikolas et al. does not have experiments or code available (and indeed we asked and they said they did not implement their algorithm). Price and Xun is one of many interesting approaches for adversarial streaming PCA, where the order of observed vectors may be adversarially chosen but the goal is to identify the top PCA subspace from the stream, whether or not that is robust to outlier data points in the stream. Our goal is not to find the top PCA subspace from the data (which would be corrupted by outliers and highly noisy points), but to estimate the ``planted'' subspace when observations have heteroscedastic additive noise. Further, neither of these methods easily generalize to subspaces of dimension greater than 1.
>
> 3. The roles of the step-size weights $w_t$ and regularization constants $(c_F, c_v)$ are explained, but practical tuning advice or an ablation study is not provided. A brief sensitivity analysis would help practitioners reproduce and adapt the method to different datasets.
> > Thank you for this request. We added extensive hyperparameter sweeps to analyze the sensitivity of our algorithm to various choices of hyperparameters. Please see plots Figures 8 and 9 in the revised appendix.

---

> > ### Comment · Reviewer_mM8x · 2025-07-17
> >
> > I understand that Price and Xun is not applicable here. Thank you for checking with Diakonikolas et al. about their implementation. It still seems an interesting point of comparison.
> >
> > Regarding generalization to subspaces of dimension more than 1, recent work (see for e.g [link](https://arxiv.org/pdf/2403.03905)) show how to use top-1 PCA algorithms and deflation to extend to top-k.
> >
> > Thank for providing the extensive hyperparameter search information.

---

### Review · Reviewer_QkZF · 2025-06-11

**Summary Of Contributions:**

In this paper, the authors look at the problem of modeling high dimensional data that lies in a lower dimensional subspace using gaussian noise with different noise levels. In particular, the authors look at the setting of streaming data with missing entries and heteroscedastic noise - in this setting, the authors consider varying levels of noise variance depending on the sample. In such an instance, the authors introduces an algorithm SHASTA-PCA that has low memory and computational overhead.

**Audience:**

Yes

**Claims And Evidence:**

Yes

**Requested Changes:**

See above

**Strengths And Weaknesses:**

The paper tackles an interesting problem but unfortunately I find the writing of the paper to be quite dense. Below, I point towards specific instances:

In Section 3, it will be better if is clearly mentioned that k is usually much smaller than d. The goal of modeling via F is to ensure that all the observed vectors lie in a low dimensional subspace.

Why do the authors assume that group memberships are known? Why is this a valid assumption in the first place?

Where is the proof of equation 3? It seems non-trivial to me at first glance

I was completely lost in Section 4 - the authors need to write sentences like the following in a significantly better fashion

Next, we take the expectation of (4) with respect to the following conditionally independent distributions obtained from Bayes’ rule and the matrix inversion lemma:

What is the matrix inversion lemma you are referring to? Cite or put the lemma in the appendix.

---

> ### Author Response · Authors · 2025-07-09
> **Response to Reviewer QkZF**
>
> The authors thank the reviewer for your helpful comments. We have revised the paper to address all of your suggested changes; please see all changes in the paper in blue text. Below we respond to your comments line-by-line:
>
> 1. In Section 3, it will be better if is clearly mentioned that k is usually much smaller than d. The goal of modeling via F is to ensure that all the observed vectors lie in a low dimensional subspace.
>
> >Thank you for this suggestion. We have clarified this in Section 3 where we first introduce our probabilistic model.
>
> 2. Why do the authors assume that group memberships are known? Why is this a valid assumption in the first place?
> >We added a sentence in Section 3. Indeed there are many applications where this is a valid assumption, especially where the instrumentation is known, e.g., sensors of different quality and noise characteristics.
>
> 3. Where is the proof of equation 3? It seems non-trivial to me at first glance.
> >The observed entries of the data vectors are distributed as
> $$
> y_{\Omega_i} \sim \mathcal{N}(0_{|\Omega_i|}, F_{\Omega_i} F_{\Omega_i}' + v_{g_i} I_{|\Omega_i|}),
> $$
> meaning the likelihood is given as follows using the Gaussian distribution
> $$
> (2\pi)^{-d/2} \det(F_{\Omega_i}F_{\Omega_i}' + v_{g_i} I_{|\Omega_i|})^{-1/2}
>     \exp \left( -\frac{1}{2} y_{\Omega_i}'(F_{\Omega_i}F_{\Omega_i}'
>         + v_{g_i} I_{|\Omega_i|})^{-1}y_{\Omega_i} \right)
>     .
> $$
> Taking log we have the log likelihood
> $$
> L_i(y_{\Omega_i}; F, v) :=     -\frac{d}{2} \ln 2\pi + \frac{1}{2} \ln\det(F_{\Omega_i}F_{\Omega_i}' + v_{g_i} I_{|\Omega_i|})^{-1} - \frac{1}{2} y_{\Omega_i}'(F_{\Omega_i}F_{\Omega_i}' + v_{g_i} I_{|\Omega_i|})^{-1}y_{\Omega_i},
> $$
> matching Eq (3) up to constant terms. We are happy to put this in the appendix if the reviewer would like. Note: we removed the trace from Eq (3); we don't need it at that point of the derivation.
>
> 4. I was completely lost in Section 4 - the authors need to write sentences like the following in a significantly better fashion
>
>     ``Next, we take the expectation of (4) with respect to the following conditionally independent distributions obtained from Bayes’ rule and the matrix inversion lemma:"
>
> >Thanks for raising this issue. We went through and tried to make things clearer, including more steps of derivation in the appendix.
>
> 5. What is the matrix inversion lemma you are referring to? Cite or put the lemma in the appendix.
> > This is fixed. We moved this discussion to an appendix and added a citation.

---

### Comment · Action_Editor_FWsk · 2025-07-16
**last week for discussion**

Dear reviewers (and authors),

The authors have submitted rebuttals, and the reviewers have 1 week left (by July 23) for their final recommendation. One of the great things about the TMLR format is that we can have a dialog, so please do not treat reviews/rebuttals as static, but rather as a starting place.

Reviewers, if you are not satisfied with the rebuttal, please leave a comment and say so! Please also take the time to read each other's reviews.  When you write your final recommendations, I won't just "average" recommendations but rather I'll look at the arguments you make, so please be prepared to describe why you give your recommendation.

I greatly appreciate the time you all spent reviewing. Please spend just a little bit more time, and then your job will be done.

-Stephen (AE for this paper)

---

### Decision · Action_Editor_FWsk · 2025-07-29

**Recommendation:** Accept as is

**Audience:**

Yes

**Audience Explanation:**

PCA is clearly of huge importance, and this paper tackles types of streaming PCA, which is a large and established subtopic. In particular, the paper studies the case when there are missing entries and heteroscedastic noise, which are relevant to many readers since these arise in practice.  This is a significant enough niche that I find enough TMLR readers will be interested.

**Claims And Evidence:**

Yes

**Claims Explanation:**

While the reviewers noted the lack of an end-to-end proof, they all thought that the empirical validation was solid, and I find these numerical experiments to be sufficient.